# WHO MATTERS MATTERS: AGENT-SPECIFIC CONSERVATIVE OFFLINE MARL

**Haosheng Chen**[1,✉], **Yun Hua**[2,✉], **Wenhao Li**[3], **Shiqin Wang**[1], **Xiangfeng Wang**[4,5,✉]

1. School of Computer Science and Technology, East China Normal University
2. Antai College of Economics and Management, Shanghai Jiao Tong University
3. School of Computer Science and Technology, Tongji University
4. Key Laboratory of Mathematics and Engineering Applications (MoE), East China Normal University
5. Shenzhen Loop Area Institute (SLAI)

hschen@stu.ecnu.edu.cn, hyyh28@sjtu.edu.cn, whli@tongji.edu.cn,
51275901137@stu.ecnu.edu.cn, xfwang@cs.ecnu.edu.cn

## ABSTRACT

Offline Multi-Agent Reinforcement Learning (MARL) enables policy learning from static datasets in multi-agent systems, eliminating the need for risky or costly environment interactions during training. A central challenge in offline MARL lies in achieving effective collaboration among heterogeneous agents under the constraints of fixed datasets, where **conservatism** is introduced to restrict behaviors to data-supported distributions. Agents with distinct roles and capabilities require individualized conservatism - yet must maintain cohesive team performance. However, existing approaches often apply uniform conservatism across all agents, leading to over-constraining critical agents and under-constraining others, which hampers effective collaboration. To address this issue, a novel framework, **OMCDA**, is proposed, where the degree of conservatism is dynamically adjusted for individual agents based on their impact on overall system performance. The framework is characterized by two key innovations: (1) A decomposed Q-function architecture is introduced to disentangle return computation from policy deviation assessment, allowing precise evaluations of each agent's contribution; and (2) An adaptive conservatism mechanism is developed to scale constraint strength according to both behavior policy divergence and the estimated importance of agents to the system. Experiments on MuJoCo and SMAC show OMCDA outperforms existing offline MARL methods, effectively balancing the flexibility and conservatism across agents while ensuring fair credit assignment and better collaboration.

## 1 INTRODUCTION

Multi-agent reinforcement learning (MARL) has gained significant traction in domains such as autonomous driving (Cao et al., 2012), collaborative robotics (Orr & Dutta, 2023), and multi-player games (Berner et al., 2019; Li et al., 2025), where agents must learn to coordinate or compete to accomplish complex objectives. Despite its successes, most MARL approaches assume agents can interact freely with the environment during training. In practice, however, this assumption often breaks down due to high interaction costs, safety concerns, or operational constraints (Wang et al., 2024). To address these limitations, Offline Reinforcement Learning (Offline RL) has emerged as a compelling alternative (Fujimoto & Gu, 2021; Kostrikov et al., 2021b; Kumar et al., 2020; Levine et al., 2020; Wu et al., 2019). Instead of relying on real-time interactions, Offline RL learns from pre-collected datasets, making it more practical for safety-critical or data-scarce environments. In the single-agent setting, Offline RL has achieved notable progress by addressing challenges such as Q-value overestimation for out-of-distribution (OOD) actions, which often lead to poor generalization. A key development in this direction is the use of *conservative* methods (Wu et al., 2019), which penalize unlikely or unsupported actions to ensure that learned policies remain close to the behavior policy. This form of **conservatism** is defined as the tendency to favor actions supported by the training data while avoiding uncertain or OOD behaviors which improves stability and robustness during offline learning (Kumar et al., 2020).

When Offline RL is extended to multi-agent settings (Offline MARL), the situation becomes even more complex. The interplay among agents introduces increased non-stationarity, and the offline dataset can exhibit more severe distributional shifts. Moreover, credit assignment—how each agent's actions contribute to overall joint performance—presents a substantial challenge (Wang & Zhan, 2023; Yang et al., 2021). Recent efforts has explored Offline MARL under the "Centralized Training and Decentralized Execution" (CTDE) framework (Lowe et al., 2017), leveraging multi-agent value decomposition combined with offline conservatism to stabilize learning.

Nevertheless, existing studies seldom consider the heterogeneity of agents in real world applications. Due to their distinct roles and interaction patterns, different agents can wield unequal influence on overall system performance (Wang et al., 2020b; Foerster et al., 2018). For instance, in a football team, strikers are often encouraged to take creative, high-risk actions to maximize scoring opportunities, while defenders must adhere to disciplined, risk-averse strategies to ensure team stability. Imposing equal conservatism on both roles would limit the striker's effectiveness and increase the defender's exposure to costly errors. This illustrates that the appropriate level of conservatism should depend on the agent's role, uncertainty, and potential impact. Consequently, a central challenge in heterogeneous Offline MARL is how to *adaptively assign conservatism* across agents based on their individual characteristics—striking a balance between safety and exploration that enhances both performance and reliability so as to promote their collaboration.

In this study, we introduce a novel offline MARL approach, **Offline MARL with Conservative Degree Allocation (OMCDA)**, which addresses the challenge of distributing conservatism among agents based on their deviations from behavior policies and their impact on system performance for heterogeneous agents in offline MARL. OMCDA decomposes the Q-function in offline MARL with regularization into two components: one for computing the return and the other for capturing policy deviations. This decomposition isolates the impact of deviations, enabling a clearer and more accurate learning process. The conservative degree of each agent is dynamically adjusted based on the effect of their deviations on the overall return, promoting a balanced influence on system performance. This dynamic allocation is integrated into the OMCDA framework, ensuring a balance between conservatism and flexibility, and consistent credit assignment to enhance teamwork.

The key contributions of this paper are as follows: **(1)** A comprehensive analysis of conservative degree allocation in heterogeneous offline MARL, exploring how varying conservative degrees affect individual agent returns and overall system performance. **(2)** The introduction of OMCDA, a novel offline MARL algorithm that dynamically adjusts each agent's conservative degree based on its impact on system performance, balancing conservatism and flexibility while ensuring consistent credit assignment and promoting collaboration. **(3)** Extensive experiments on diverse datasets, including multi-agent MuJoCo (de Witt et al., 2020) and the StarCraft Multi-Agent Challenge (SMAC) (Samvelyan et al., 2019), showing that OMCDA consistently outperforms existing methods across different environments and datasets.

## 2 PRELIMINARIES

We consider a MARL problem following (Wang et al., 2024) where the environment is modeled as a multi-agent Partially Observable Markov Decision Process (Boutilier, 1996), defined by the tuple: $G = \langle \mathcal{S}, \mathcal{A}, P, r, \mathcal{Z}, \mathcal{O}, n, \gamma \rangle$. $s \in \mathcal{S}$ is the true state of the environment. $\mathcal{A}$ denotes the action set for each of the $n$ agents. At every time step, each agent $i \in \{1, 2, \ldots, n\}$ chooses an action $a_i \in \mathcal{A}$, forming a joint action $\boldsymbol{a} = (a_1, a_2, \ldots, a_n) \in \mathcal{A}^n$. It causes a transition to the next state $s'$ in the environment according to the transition dynamics $P(s'|s, \boldsymbol{a}) : \mathcal{S} \times \mathcal{A}^n \times \mathcal{S} \to [0, 1]$. All agents share the same global reward function $r(s, \boldsymbol{a}) : \mathcal{S} \times \mathcal{A}^n \to \mathbb{R}$. $\gamma \in [0, 1)$ is a discount factor. In the partially observable environment, each agent draws an observation $o_i \in \mathcal{O}$ at each step from the observation function $\mathcal{Z}(s, i) : \mathcal{S} \times N \to \mathcal{O}$. The objective of the team is to learn a set of policies $\pi = (\pi_1, \pi_2, \ldots, \pi_n)$ that collectively maximize the expected discounted cumulative reward of the entire system. In the offline setting, agents do not interact with the environment directly but instead learn policies from a static dataset $\mathcal{D}$ containing state-action-reward tuples. The challenge lies in learning effective policies without additional environment interactions.

**CTDE Framework**   The Centralized Training with Decentralized Execution (CTDE) framework is widely used in cooperative multi-agent reinforcement learning (MARL) (Oliehoek et al., 2008). In

CTDE, agents are trained centrally with global information, enabling coordinated policy learning (Lowe et al., 2017). During execution, they act based on decentralized local observations, ensuring scalability in real-world settings. A key approach in this framework is value decomposition (Rashid et al., 2020; Sunehag et al., 2017; Wang et al., 2020a), where the global value function is factorized into local components for each agent. Algorithms such as QMIX (Rashid et al., 2020) and VDN (Sunehag et al., 2017) employ monotonic value decomposition for scalable multi-agent learning. This framework has been adopted in offline MARL (Pan et al., 2022; Yang et al., 2021), with the global-local Q-value relationship:

$$Q_{tot}(\mathbf{o}, \mathbf{a}) = \sum_i w_i(\mathbf{o})Q_i(o_i, a_i) + b(\mathbf{o}), w_i \geq 0, \quad \forall i = 1 \cdots n. \tag{1}$$

where $w_i(\mathbf{o})$, $b(\mathbf{o})$ are local function weights/bias and $a_i$, $o_i$ are agent actions/observations.

**Offline MARL with Policy regularization**  Policy regularization constrains policy learning to remain close to the behavior policy (Xu et al., 2023), preventing out-of-distribution actions that could degrade performance. Several offline RL algorithms (Wu et al., 2019; Xu et al., 2023) use this approach to mitigate distributional shift. For example, BRAC (Wu et al., 2019) regularizes the actor's policy to stay close to the behavior policy while optimizing the critic with the standard value function update. In offline MARL settings, this regularized method can be extended, with the objective written as:

$$\max_\pi E\left[\sum_{t=0}^\infty \gamma^t r_t\right], \text{s.t.} \quad E_{a \sim \pi}\left[f\left(\pi\left(a_t \mid o_t\right), \pi_b\left(a_t \mid o_t\right)\right)\right] \leq \epsilon. \tag{2}$$

Here $\pi = (\pi_1, \ldots, \pi_N)$, $f$ is a divergence term that quantifies how far the policy deviates from the behavior policy $\pi_b$, $a_t$ and $o_t$ are the action and state at timestep $t$, while $\epsilon$ is the constraint of $f$. We then convert the constrained optimization problem above into an unconstrained one using a Lagrangian relaxation by introducing a penalty hyperparameter $\alpha$:

$$\max_\pi \mathbb{E}\left[\sum_{t=0}^\infty \gamma^t\left(r_t - \alpha \cdot f\left(\pi\left(a_t \mid o_t\right), \pi_b\left(a_t \mid o_t\right)\right)\right)\right]. \tag{3}$$

In this paper, we use Kullback-Leibler (KL) divergence(Pérez-Cruz, 2008) expressed as $D_{\text{KL}}$ to constrain the learning policy and behaviour policy, while the Q-function can be formulated as:

$$Q\left(o, a\right) = \mathbb{E}\left[\sum_{t=0}^\infty \gamma^t\left(r_t - \alpha \cdot D_{\text{KL}}(\pi_t \parallel \pi_b)\right)\right], \tag{4}$$

where $\pi_t$ and $r_t$ are the policy and reward at timestep $t$. To address the issue of conservative degree allocation, we provide different levels of conservatism to agents in offline MARL by assigning each agent $i$ an individual **conservative degree** $d_i$, which defines the permissible range of deviation from its behavior policy. The problem is then reformulated as the following:

$$\max_\pi E\left[\sum_{t=0}^\infty \gamma^t r_t\right], \text{ s.t. } E_{a \sim \pi}\left[\log \frac{\pi_t^i(a_t^i \mid o_t^i)}{\pi_b(a_t^i \mid o_t^i)}\right] \leq d_i, \quad \sum_i d_i = d_{tot}, \forall i = 1 \cdots n. \tag{5}$$

Where $d_i$ is the local conservative degree, and $d_{tot}$ is the global conservative degree which is a fixed value. A deeper analysis of Eq. (5), which reveals the origin of the deviation term in Eq. (12), is provided in Appendix E.5. Then similar to the process of transitioning from Eq. (2) to Eq. (3), we can convert Eq. (5) and assign a **conservatism level** to each agent $i$, denoted as $\alpha_i$, while the current Q-function can be formulated as:

$$Q\left(o, a\right) = \mathbb{E}\left[\sum_{t=0}^\infty \gamma^t\left(r_t - \sum_i \alpha_i \cdot D_{\text{KL}}(\pi_t^i \parallel \pi_b)\right)\right]. \tag{6}$$

In the next section, OMCDA is introduced, built upon the decomposition of the Q-function and dynamic conservative degree allocation. We will demonstrate how this framework addresses the challenges of conservative degree allocation and emphasize it's advantages in offline MARL systems.

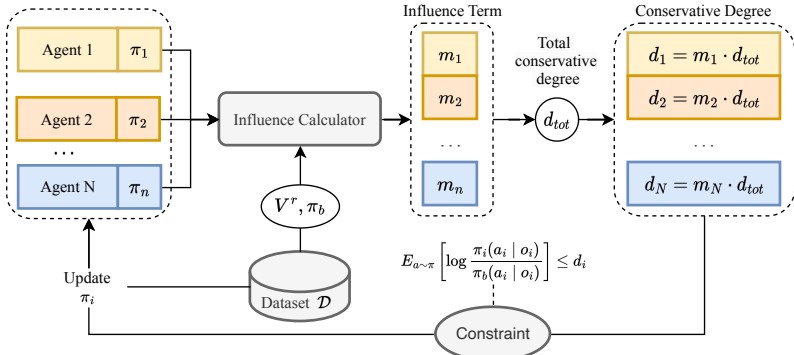

Figure 1: Overview of dynamic conservative degree allocation framework in OMCDA. 1) The influence calculator takes the policy from each agent, along with the return-based state-value function $V^r$ and the behavior policy $\pi_b$ derived from the data in dataset, as input to generate the influence term for each agent on the system. 2) Each agent's conservative degree is then allocated from the total conservative degree based on the influence term. 3) Finally, the conservative degree is integrated as a network update constraint, enabling dynamic allocation while ensuring consistent credit assignment.

## 3 OMCDA

In this section, we present OMCDA for dynamic conservative degree allocation in offline MARL. First, we motivate the problem through a simple example, then decompose the Q-function to quantify individual agent contributions. We develop an adaptive mechanism that adjusts each agent's conservatism according to its impact on system rewards. OMCDA ensures conservatism levels align with agents' influence on returns, enhancing performance while preserving consistent credit assignment.

**Conservative Degree Allocation in Offline MARL**   In offline MARL, agents' influences on the system are not uniform. To fully leverage these influences and improve system performance, dynamic conservative degree allocation is necessary. This approach allows high-impact agents to make larger deviations, enhancing their contribution to the overall performance. To better illustrate this issue, we present a 2-player toy example in Table 1.

In this game, we employ a **mixed strategy**. As a cooperative team, individual players cannot access their personal rewards directly; instead, they observe the team's expected reward $r_{team}$ which is :

$$\sum_{a_1 \in \{A,B\}} \sum_{a_2 \in \{A,B\}} \pi_1(a_1)\pi_2(a_2)r(a_1, a_2). \tag{7}$$

The game features two possible actions ($A$ and $B$) for each player. Consider both players following a uniform behavior policy $\pi_b = (0.5, 0.5)$. The offline dataset $\mathcal{D}$ is collected under $\pi_b$, containing policy pairs with their corresponding team reward $r_{\text{team}}$: $\mathcal{D} = \{(\pi_1 = (0.5, 0.5), \pi_2 = (0.5, 0.5), r_{\text{team}})\}$. Clearly, Player 1 achieves higher rewards and greater influence on team performance than Player 2, justifying greater allowance for policy deviation.

We quantify deviation of agent $i$'s policy from behavior policy through Manhattan distance (Chiu et al., 2016) $\Delta_i = \sum_{a \in \{A,B\}} |\pi_i(a) - \pi_{\text{behavior}}(a)|$.

To align with traditional offline methods, the total conservative degree $\Delta_{\text{total}}$ for the entire system is set to 0.4. Under uniform conservative allocation, both players share the total deviation equally: $\Delta_1 = \Delta_2 = 0.2$. After training, the

Table 1: Payoff matrix for the 2-player example.

| Player 1/2 | A | B |
|---|---|---|
| **A** | $3,1$ | $3,1$ |
| **B** | $2,1$ | $2,1$ |

optimal strategies for both players are $(0.6, 0.4)$, increasing team's expected reward by $0.1$. Under dynamic allocation, Player 1 receives a larger deviation: $\Delta_1 = 0.3$, reflecting its higher impact, while $\Delta_2 = 0.1$. Consequently, Player 1 learns a more aggressive strategy $(0.65, 0.35)$, while Player 2

remains near the behavior policy $(0.55, 0.45)$. The dynamic allocation improves team's expected reward more by $0.15$, demonstrating its effectiveness in coordinating heterogeneous agents.

**Decomposition Framework** We now present the decomposition framework for value functions, aiming to assign different conservative degrees to each agent, as described in Eq. (5). To achieve this, it's crucial to understand how an agent's deviation from the behavior policy affects the overall return. In offline RL with regularization, both the Q-function and value function contain entangled return and constraint components (Eq. 4), complicating the measurement of an agent's influence on the return. Inspired by BOPAH (Lee et al., 2020), our framework disentangles these components by decomposing the Q-function and value function into two parts: one that computes the return and another that accounts for the deviation constraint. In our framework, the original Q-function in offline RL with regularization in Eq.(4) can be written as:

$$Q\left(o, a\right) = Q^r\left(o, a\right) + \alpha \cdot Q^c\left(o, a\right), \ Q^r := \mathbb{E}\left[\sum_{t=0}^{\infty} \gamma^t r_t\right], \ Q^c := \mathbb{E}\left[-\sum_{t=1}^{\infty} \gamma^t D_{\text{KL}}(\pi_t \parallel \pi_b)\right]. \quad (8)$$

In this definition, $Q^r\left(o, a\right)$ calculates the return, and $Q^c\left(o, a\right)$ captures the deviation from the behavior policy. Similar to these two Q-functions, the decomposition of V-function can also be obtained as:

$$V\left(o, a\right) = V^r\left(o, a\right) + \alpha \cdot V^c\left(o, a\right). \quad (9)$$

Then with the current Q-function and V-function, the corresponding Bellman backup operators is formulated as:

$$\left(\mathcal{T}_f^\pi\right) Q^r(o, a) := r(o, a) + \gamma \mathbb{E}_{o'|o,a}\left[V^r\left(o'\right)\right], \ \left(\mathcal{T}_f^\pi\right) Q^c(o, a) := \gamma \mathbb{E}_{o'|o,a}\left[V^c\left(o'\right)\right], \quad (10)$$

where V-functions is written similar to SAC (Haarnoja et al., 2018) as:

$$V^r(o) = \mathbb{E}_{a\sim\pi}\left[Q^r(o, a)\right], \ V^c(o) = \mathbb{E}_{a\sim\pi}\left[Q^c(o, a) - \log\left(\frac{\pi(a \mid o)}{\pi_b(a \mid o)}\right)\right]. \quad (11)$$

By decoupling the Q-function into separate return and deviation components, we isolate return calculation from conservatism enforcement. This enables precise assessment of each agent's influence on cumulative returns, free from conservatism constraint interference. This approach proves particularly crucial in offline MARL, where individual actions affect the joint return. When extending to multi-agent case, according to Eq. (6) and the definitions in Eq. (8), the global Q-function (a detailed analysis of the relationship between Eq. (5) and the deviation term in Eq. (12) is provided in Appendix E.5) can be derived within the QMIX framework (Rashid et al., 2020) as follows:

$$Q_{tot}\left(o, a\right) = Q_{tot}^r\left(o, a\right) + \sum_{i=1}^{N} \alpha_i \cdot Q^{c,i}\left(o, a\right), \quad (12)$$

where

$$Q_{tot}^r(\boldsymbol{o}, \boldsymbol{a}) = \sum_i w_i^r(\mathbf{o})Q_i^r(o_i, a_i) + b^r(\mathbf{o}), \quad (13)$$

$$Q^{c,i}(\boldsymbol{o}, \boldsymbol{a}) = \sum_j w_j^{c,i}(\boldsymbol{o})Q_j^c\left(o_j, a_j\right) + b^{c,i}(\boldsymbol{o}). \quad (14)$$

In Eq. (13), $Q_{tot}^r$ represents the global return information, which is distributed to individual agents through value decomposition, with $w^r$ and $b^r$ as the weight and bias parameters for each agent's local return function $Q_i^r$. Eq. (14) defines agent $i$'s conservatism value function $Q^{c,i}(\boldsymbol{o}, \boldsymbol{a})$, computed as a weighted sum over all agents' conservatism values $Q_j^c(o_j, a_j)$. Here, $w_j^{c,i}(\boldsymbol{o})$ denotes the observation-dependent weight for agent $j$'s contribution to agent $i$'s conservatism, while $b^{c,i}(\boldsymbol{o})$ serves as an adaptive bias term. The decomposition of the $V$-function is derived in the same manner as the $Q$-function. The decomposed forms of $V_{tot}^r$ and $V^{c,i}$ are expressed as follows:

$$V_{tot}^r(\mathbf{o}) = \sum_i w_i^r(\mathbf{o})V_i^r(o_i) + b^r(\mathbf{o}), \quad (15)$$

$$V^{c,i}(\boldsymbol{o}) = \sum_j w_j^{c,i}(\boldsymbol{o})V_j^c(o_j) + b^{c,i}(\boldsymbol{o}). \quad (16)$$

With the decomposition framework, each agent can balance both individual and global constraints effectively, while also more accurately assessing both its own and the overall system's return.

**Dynamic Conservative Degree Allocation for Agents in Offline MARL Setting** After we get our decomposition framework, since the goal to maximum the return is equal to maximum global return-based state-value function $V_{tot}^r$. The maximum term in Eq.(5) can be changed into $\max_\pi \mathbb{E}\left[V_{tot}^r(o)\right]$.

Next, we illustrate the approach to develop a dynamic adaptation method (shown in Figure. 1) that adjusts the conservative degrees for agents dynamically. Let us take another look at the constraint term in Eq.(5) where we want to adaptively assign a conservative degree $d_i$ to each agent. Given the total degree $d_{tot}$, an efficient strategy is to allocate it based on the influence of the agents on the system. Hence, we propose an influence term $\boldsymbol{m_i}$ for each agent $i$, and $d_i$ can be obtained as:

$$d_i = \boldsymbol{m_i} \cdot d_{tot}. \tag{17}$$

As shown in Table 1, in offline MARL settings, an agent's influence on the system determines the sensitivity of the system to its behavioral policy deviations. Thus, we quantify each agent's influence as the impact of its policy deviation on the collective return $V_{tot}^r$. Taking the expression of $V_{tot}^r$ in Eq.(11) into account, the influence can be derived by computing the partial derivative of the return-based value function $V_{tot}^r$ with respect to the KL divergence between the agent's current policy $\pi_i$ and its own behavior policy $\pi_b^i$ following:

$$\boldsymbol{m_i} = \frac{\partial V_{tot}^r(o)}{\partial D_{\mathrm{KL}}(\pi_i \parallel \pi_b^i)}. \tag{18}$$

In practice, to facilitate computation, the chain rule is applied to break down complex dependencies between $V_{tot}^r(o)$ and the KL divergence $D_{\mathrm{KL}}(\pi^i \parallel \pi_b^i)$, enabling efficient influence computation (further details are in Appendix E.2):

$$\boldsymbol{m_i} = \frac{\partial V_{tot}^r(o)}{\partial \pi_i} \left(\frac{\partial D_{\mathrm{KL}}(\pi_i \parallel \pi_b^i)}{\partial \pi_i}\right)^{-1}. \tag{19}$$

The first term in Eq.(19) captures the strategy change's system impact, while the second term acts as a constraint, measuring the agent's deviation from its behavior policy. Since $V_{tot}^r$ isolates the conservatism term from $V_{tot}$, it directly quantifies how policy deviations affect system returns. This reveals the relationship between an agent's policy deviation and its return impact. Eq.(18) dynamically determines each agent's conservatism constraint, measuring system return sensitivity to policy deviations. A larger derivative indicates greater positive return impact, permitting more flexible $d_i$; smaller derivatives warrant stricter constraints to mitigate risk. Due to $\sum_i d_i = d_{tot}$ in Eq.(5), to determine the appropriate conservative degree $d_i$ for each agent, we adopt a softmax function to normalize the weights across all agents:

$$\boldsymbol{m} = [\boldsymbol{m_1}, \cdots, \boldsymbol{m_N}] = \mathrm{Softmax}\left[\mathbb{E}\left[\frac{\partial V_{tot}^r(o)}{\partial D_{\mathrm{KL}}(\pi_1 \parallel \pi_b^1)}\right], \cdots, \mathbb{E}\left[\frac{\partial V_{tot}^r(o)}{\partial D_{\mathrm{KL}}(\pi_N \parallel \pi_b^N)}\right]\right]. \tag{20}$$

After obtaining $m_i$, each $d_i$ can be derived using Eq.(17). The conservatism level $\alpha_i$ introduced in Eq.(12) can be updated according to following objective:

$$\min_{\alpha_i}\left(\alpha_i \cdot d_i - \alpha_i \cdot D_{\mathrm{KL}}\left(\pi_i \parallel \pi_b^i\right)\right). \tag{21}$$

With the conservatism levels $\alpha_i$ obtained for each agent, we apply these dynamic adjustments to the offline MARL environment. We begin by deriving the optimal global policy in the offline MARL setting in Proposition 3.1.

**Proposition 3.1.** *In an offline MARL setting, the optimal global policy $\pi_{tot}^*(a \mid o)$ is given by Eq. (4) and is formally expressed as follows:*

$$\pi_{tot}^*(a \mid o) = \pi_b(a \mid o) \cdot \exp\left(\frac{1}{\alpha}\left(Q^*(o, a) - V^*(o)\right)\right), \tag{22}$$

*where $Q^*(o, a)$ is optimal action-value function, $V^*(o)$ is optimal value function, and a global conservatism level $\alpha$ is assumed that controls the overall deviation from the behavior policy.*

The proof is based on the principles of soft Q-learning(Haarnoja et al., 2018) and we extend it to offline MARL context. Then, we aim to derive the local optimal policy from the global optimal policy in Eq.(22) by applying the individual conservatism level $\alpha_i$ for each agent in Proposition 3.2, and demonstrate its validity in Theorem 3.3.

**Proposition 3.2.** *Joint policy $\pi_{tot}$ is decomposed into product of individual agent policies $\pi_i$ as:*

$$\pi_{tot}(a \mid o) = \prod_{i=1}^{N} \pi_i(a_i \mid o_i).$$

*Based on decomposition in Eq.(12) - (16), the optimal policy $\pi_i^*(a_i \mid o_i)$ for each agent is given by:*

$$\pi_i^*(a_i \mid o_i) = \pi_b(a_i \mid o_i) \cdot \exp\left( \frac{w_i^r(\mathbf{o})}{\alpha_i} \left( Q_i^{r^*}(o_i, a_i) - V_i^{r^*}(o_i) \right) + \left( Q^{c,i^*}(\mathbf{o}, a) - V^{c,i^*}(\mathbf{o}) \right) \right),$$
(23)

*where $\alpha_i$ controls the conservatism level of the agent's policy, and $\pi_i^*$ denotes the **optimal policy** that satisfies Eq.(5).*

**Theorem 3.3.** *Given Eq.(23), the optimal policy for each agent is derived, and consistency between the local optimal policies $\pi_i^*$ and the global optimal policy $\pi_{tot}^*$ is guaranteed. This consistency holds for individual $\alpha_i$ assignments across agents.*

The optimal $\pi_i^*$ is then used to update each agent's conservatism-based value function $V_i^c$. It should be noted that each local policy needs to satisfy $\sum_{a_i \sim \pi_i} \pi_i^*(a_i \mid o_i) = 1$. Therefore, according to Eq.(23), the following formula can be obtained:

$$\mathbb{E}_{a_i \sim \pi_b}\left[ \exp\left( \frac{w_i^r(\mathbf{o})}{\alpha_i} \left( Q_i^{r^*}(o_i, a_i) - V_i^{r^*}(o_i) \right) + \left( Q^{c,i^*}(\mathbf{o}, a) - V^{c,i^*}(\mathbf{o}) \right) \right) \right] = 1. \qquad (24)$$

**Proposition 3.4.** *From Eq.(24), each agent's conservatism-based value function $V_i^c$ is updated through the following optimization:*

$$\begin{aligned}
\min_{V_i^c} \mathbb{E}_{(o_i, a_i) \sim \mathcal{D}} &\left[ \exp\left( \frac{w_i^r(\boldsymbol{o})}{\alpha_i} \left( Q_i^r(o_i, a_i) - V_i^r(o_i) \right) \right. \right. \\
&\left. + \left( Q^{c,i}(o, a) - V^{c,i}(o) \right) \right) + \frac{w_i^r(\boldsymbol{o}) V_i^r(o_i) + \alpha_i V^{c,i}(o)}{\alpha_i} \right].
\end{aligned}$$
(25)

The proofs of Proposition 3.1, Proposition 3.4 and Theorem 3.3 are provided in Appendix C.

Dynamic conservatism has now been incorporated into MARL frameworks, enabling agents to optimize their behavior in offline settings through adaptive balancing between conservatism and policy deviation—with each agent's contribution weighted by its measured impact on collective system performance. The algorithm and additional explanation of OMCDA is in Appendix E.1.

**Comparison with prior works**  Prior works including FOP (Zhang et al., 2021), ADER (Kim & Sung, 2023), and CFCQL (Shao et al., 2024) have investigated adaptive approaches in MARL. While FOP and ADER are online methods that employ dynamic entropy regularization similar to OMCDA, their adaptive mechanisms are confined to policy updates, applying either global uniform constraints or no constraints to $Q/V$ function updates-an approach that fails to address per-agent constraint allocation for heterogeneous agents, which is crucial for mitigating OOD issues in offline MARL. In contrast, OMCDA uniquely enables dynamic conservatism allocation for both policies and value functions, ensuring optimal updates in offline settings. Offline method CFCQL determines conservatism by behavior policy deviation, while OMCDA considers each agent's impact on system performance. This allows OMCDA to balance conservatism and flexibility, optimizing performance.

## 4 EXPERIMENT

In this section, we conduct experiments to: **(1)** evaluate OMCDA's performance, **(2)** demonstrate its effectiveness in dynamic conservative degree allocation, and **(3)** analyze key components and choices of total conservative degrees. Further ablation details are in Appendix F.

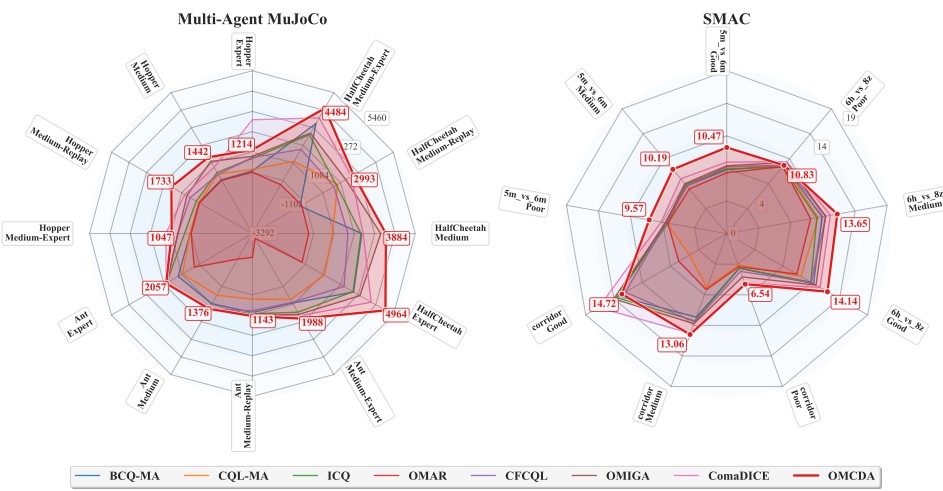

Figure 2: The average returns for the offline Multi-Agent MuJoCo and SMAC

**Offline Multi-Agent Datasets**    We select **Multi-Agent MuJoCo**(de Witt et al., 2020) and the **StarCraft Multi-Agent Challenge (SMAC)**(Samvelyan et al., 2019) as our experiment environments. Multi-Agent MuJoCo, a benchmark for continuous multi-agent robotic control, is built on the MuJoCo environment. The Multi-Agent MuJoCo dataset we use was collected using the HAPPO(Kuba et al., 2021) algorithm by (Wang et al., 2024) which contains four quality levels: expert, medium, medium-replay and medium-expert. The second environment, SMAC, is a widely-used benchmark for evaluating cooperative MARL methods. The offline SMAC dataset is collected by (Meng et al., 2021), using online-trained MAPPO(Kuba et al., 2021) agents. This is the largest publicly available dataset for SMAC and includes three quality levels: good, medium, and poor. We focus on three representative battle maps in our experiments: one hard map (5m_vs_6m) and two super hard maps (6h_vs_8z and corridor). We initialize the **behavior policy** $\pi_b$ through behavior cloning (Michie et al., 1990) using the offline dataset. Further details on these datasets are provided in Appendix D.

**Baselines and Comparative Evaluation**    We compare our approach with seven offline MARL algorithms: The multi-agent versions of BCQ(Fujimoto et al., 2019) and CQL(Kostrikov et al., 2021b) (referred to as BCQ-MA and CQL-MA), ICQ(Yang et al., 2021), OMAR(Pan et al., 2022), CFCQL(Shao et al., 2024), OMIGA(Wang et al., 2024), and ComaDICE(Bui et al., 2024). Both BCQ-MA and CQL-MA utilize a linear weighted value decomposition for the multi-agent setting, similar to Eq. (1). Hyperparameters used in our experiments are provided in Appendix E.4. Figure 2 presents returns for the offline Multi-Agent MuJoCo and SMAC tasks with 5 random seeds. Detailed analysis of the results and the mean and standard deviation of returns are in Appendix E.3.

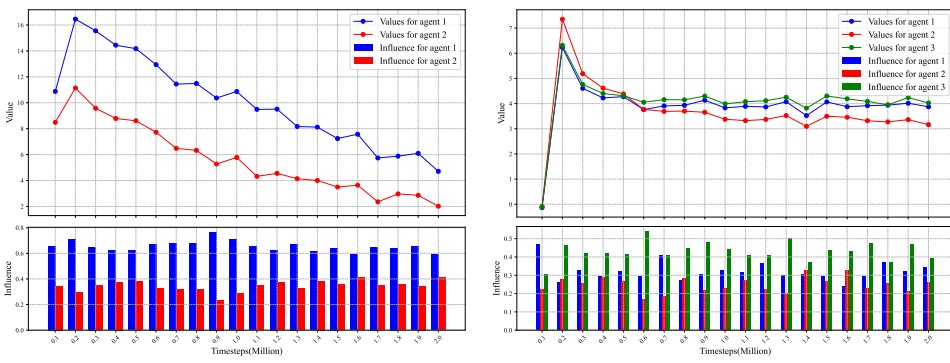

Figure 3: Analysis on the influence term on Ant(**left**) and Hopper(**right**)

**Analysis on the Influence Term**    In OMCDA, the influence of each agent on the system is the core metric for allocating conservatism levels. We conduct experiments to analyze the relationship between the computed influence of each agent and its corresponding return. The results in Figure. 3 demonstrate that agents with higher $V_i^r$, representing higher individual returns, tend to be allocated more influence, enabling them to have a stronger impact on system performance. This proportional allocation allows high-return agents to further contribute to global objectives and optimize overall behaviour. By adjusting conservatism levels properly, OMCDA enhances individual performance and maximizes collective return, promoting balanced and efficient learning across agents.

**Analysis on the Components of OMCDA**    To analyze conservative degree allocation and the impact of Q-function decomposition in OMCDA, we conduct three ablation studies: OMCDA-w/o-CDA, OMCDA-w/o-dq, and OMCDA-rd. In OMCDA-w/o-CDA, all agents share the same conservative degree $d_i$ without allocation. In OMCDA-w/o-dq, dynamic allocation remains but Q-function decomposition is removed, entangling return optimization with deviation handling. OMCDA-rd assigns each agent a random $d_i$, used to evaluate the role of strategic assignment. Experiments on HalfCheetah and 6h_vs_8z in the Multi-agent MuJoCo and SMAC environments show that OMCDA consistently outperforms all ablated versions (Figure. 4a-b). Lacking dynamic allocation, OMCDA-w/o-CDA causes imbalance and degraded performance. OMCDA-w/o-dq weakens learning as objectives become entangled, while OMCDA-rd performs worse since random $d_i$ ignores agents' distinct impact. These results confirm that dynamic allocation and Q-function decomposition are essential for collaboration and efficiency in offline multi-agent environments, while strategic assignment of conservatism is crucial for optimal system performance.

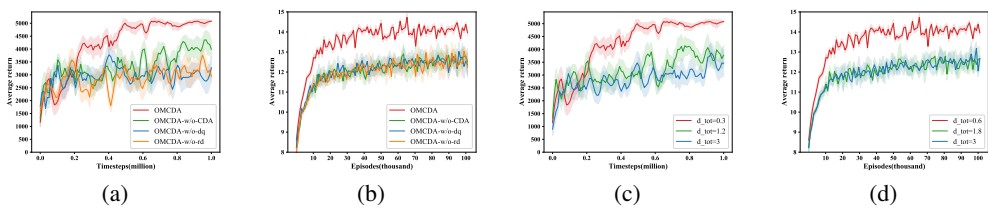

|        (a)        |        (b)        |        (c)        |        (d)        |

Figure 4: Analyses and ablations of OMCDA. We analyze the effect of model components **(a-b)** and total conservative degree **(c-d)** across HalfCheetah from MA-MuJoCo and 6h_vs_8z from SMAC.

**Analysis on the Total Conservative Degree**    The total conservative degree $d_{tot}$ controls how much the system may deviate from the behavior policy. It sets the permissible deviation for the entire system, ensuring agents do not diverge excessively. In experiments on HalfCheetah and 6h_vs_8z, based on high-quality datasets, a smaller $d_{tot}$ outperforms other settings. This is because in such environments it is essential for policies to stay closer to the behavior policy for stable performance. Meanwhile, dynamic allocation of $d_i$ allows agents with significant impact on returns some flexibility to deviate, while requiring others to remain conservative and adhere closely to the behavior policy. The results in Figure. 4(c-d) show that adjusting $d_{tot}$ improves overall performance, allowing influential agents beneficial deviations while maintaining system stability.

## 5    CONCLUSION

In conclusion, a novel offline MARL framework OMCDA is introduced to tackle the challenge of conservative degree allocation. OMCDA decomposes the Q-function in offline MARL with regularization into two components: one for computing the return and another for capturing deviations from the behavior policy. It dynamically adjusts each agent's conservative degree based on their influence on the overall system's performance, ensuring coherent credit assignment and robust performance throughout the learning process. Meanwhile, extensive experiments demonstrate that OMCDA consistently outperforms existing offline MARL methods across various environments. Our future works aim to enhance OMCDA by developing adaptive mechanisms that reduce sensitivity to total conservative degree selection, and lower the computational complexity.

## 6 Acknowledgements

Yun Hua is supported by the Shanghai Post-doctoral Excellence Program (2025251). Wenhao Li is supported by the NSFC (62406270) and the STCSM Shanghai Rising-Star Program (24YF2748800). Xiangfeng Wang is supported by the NSFC (62231019) and SHEITC (2025-GZL-RGZN-BTBX-01004).

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

## A  USE OF LARGE LANGUAGE MODELS (LLMs)

We employed the large language model as an auxiliary tool during manuscript preparation. Specifically, it was used to refine language for grammar and clarity, and to generate illustrative (non-experimental) figures based on prompts we provided. All research ideas, methods, experiments, analyses, and conclusions were developed by the authors.

## B  RELATED WORK

### B.1  OFFLINE REINFORCEMENT LEARNING

Offline reinforcement learning must address distributional shift (Kumar et al., 2019), which occurs when policies encounter out-of-distribution (OOD) states or actions (Fujimoto et al., 2019), leading to exploitation errors and poor performance due to inaccurate value estimates on OOD actions.

To mitigate this, policy constraint methods (Cheng et al., 2024; Fujimoto et al., 2019; Xu et al., 2021) aim to keep the learned policy close to the behavior policy, reducing deviations from the training data. Value regularization techniques (Kostrikov et al., 2021a) (Kumar et al., 2020) penalize OOD value estimates, while uncertainty-based (Bai et al., 2022; Wu et al., 2021) and model-based (Yu et al., 2020; Zhan et al., 2021) approaches focus on penalizing actions in uncertain or sparse regions. Recently, in-sample learning methods (Brandfonbrener et al., 2021; Kostrikov et al., 2021b; Peng et al., 2019; Xu et al., 2023) have focused on learning within the support of the offline data, avoiding OOD evaluation and improving stability. Our approach integrates multi-agent value decomposition into this paradigm, ensuring more stable and coordinated policy learning in multi-agent settings.

### B.2  MULTI-AGENT REINFORCEMENT LEARNING

A key challenge in MARL is the joint action space (Hernandez-Leal et al., 2019), which grows exponentially with the number of agents, making it difficult to find optimal policies. The Centralized Training with Decentralized Execution (CTDE) framework (Kraemer & Banerjee, 2016; Oliehoek et al., 2008; Sunehag et al., 2017) addresses this by training agents centrally with global information, while they execute based on decentralized policies using only local observations.

Recent offline MARL approaches (Jiang & Lu, 2023; Pan et al., 2022; Shao et al., 2024; Wang et al., 2024; 2023; Yang et al., 2021; Zhu et al., 2025; Liu et al., 2024; Bui et al., 2024), extend online MARL methods with regularization to avoid OOD actions. For instance, ICQ (Yang et al., 2021) uses importance sampling for local policy constraints, while OMAR (Pan et al., 2022) adapts conservative Q-learning. In contrast to value decomposition methods, which adhere to the IGM principle, ComaDice (Bui et al., 2024) employs stationary distribution shift regularization to combat the distribution shift issue. MADiff (Zhu et al., 2025) uses an attention-based diffusion model to effectively model agent collaboration. InSPO (Liu et al., 2024) sequentially optimizes agent policies in an in-sample manner. MACCA (Wang et al., 2023) and OMIGA (Wang et al., 2024) introduce global-to-local value regularization. However, these methods apply a fixed conservatism level for each agent, which can be inefficient. Although CFCQL (Shao et al., 2024) incorporates conservative value estimation, it fails to account for the heterogeneous impact of individual agents on overall system performance. Our algorithm addresses the above problems by dynamically adjusting conservative degree based on each agent's impact on the system.

## C  PROOFS

**Proposition.**3.1 *In an offline MARL setting, the optimal global policy $\pi^*_{tot}(a \mid o)$ is given by Eq. (4) and is formally expressed as follows:*

$$\pi^*_{tot}(a \mid o) = \pi_b(a \mid o) \cdot \exp\left(\frac{1}{\alpha}\left(Q^*(o,a) - V^*(o)\right)\right), \tag{26}$$

*where $Q^*(o,a)$ is optimal action-value function, $V^*(o)$ is the optimal value function, and we assume there is a global conservatism level $\alpha$ that controls the overall deviation from the behavior policy.*

*Proof.* The proof follows (Yang et al., 2021) and is the extension of SAC(Haarnoja et al., 2018) into offline multi-agent setting.

Let us return to the definition of offline MARL with regularization, we start with the original form:

$$\max_{\pi_{tot}} E_{a \sim \pi_{tot}} [Q_{tot}(o, a)],$$

$$\text{s.t. } D_{KL}(\pi_{tot} \| \pi_b) \leq \epsilon, \quad \sum_a \pi_{tot}(a \mid o) = 1. \tag{27}$$

We find that the objective is a linear function of the decision variables $\pi_{tot}$ and all constraints are convex functions. Thus Eq. (27) is a convex optimization problem. The Lagrangian equation is:

$$\mathcal{L}(\pi_{tot}, \alpha, \lambda) = \mathbb{E}_{a \sim \pi_{tot}} [Q_{tot}(o, a)] + \alpha \left( \epsilon - D_{\mathrm{KL}}(\pi_{tot} \| \pi_b) \right)$$

$$+ \lambda \left( 1 - \sum_a \pi_{tot}(a \mid o) \right), \tag{28}$$

where $\alpha$ denotes the Lagrangian coefficient which is a global conservatism level that controls the overall deviation from the behavior policy. Then we can get the following formula:

$$\frac{\partial \mathcal{L}}{\partial \pi_{tot}} = Q_{tot}(o, a) - \alpha \left( 1 + \log \left( \frac{\pi_{tot}(a \mid o)}{\pi_b(a \mid o)} \right) \right) - \lambda. \tag{29}$$

Setting $\frac{\partial \mathcal{L}}{\partial \pi_{tot}}$ to zero, then:

$$Q_{tot}(o, a) - \alpha \left( 1 + \log \left( \frac{\pi_{tot}(a \mid o)}{\pi_b(a \mid o)} \right) \right) - \lambda = 0, \tag{30}$$

$$Q_{tot}(o, a) = \alpha \left( 1 + \log \left( \frac{\pi_{tot}(a \mid o)}{\pi_b(a \mid o)} \right) \right) + \lambda, \tag{31}$$

$$\frac{Q_{tot}(o, a)}{\alpha} - \frac{\lambda}{\alpha} - 1 = \log \left( \frac{\pi_{tot}(a \mid o)}{\pi_b(a \mid o)} \right), \tag{32}$$

$$\pi_{tot}(a \mid o) = \pi_b(a \mid o) \exp \left( \frac{Q_{tot}(o, a)}{\alpha} - 1 - \frac{\lambda}{\alpha} \right). \tag{33}$$

The optimal policy is expressed similar to Eq.(33) while adding optimal symbol to all functions, which is $\pi$ to $\pi^*$. Integrating Eq.(33) with optimal symbol into the expression of optimal V-function in offline MARL with regularization, we can get:

$$V_{tot}^*(o) = \sum_a \pi_{tot}^*(a \mid o) \left( Q_{tot}^*(o, a) - \alpha \log \left( \frac{\pi_{tot}^*(a \mid o)}{\pi_b(a \mid o)} \right) \right)$$

$$= \sum_a \pi_{\mathrm{tot}}^*(a|o) (\lambda^* + \alpha) \tag{34}$$

$$= \lambda^* + \alpha.$$

Through Eq.(33) with optimal symbol and Eq.(34), we can finally obtain the optimal global policy $\pi_{tot}^*(a \mid o)$:

$$\pi_{tot}^*(a \mid o) = \pi_b(a \mid o) \cdot \exp \left( \frac{1}{\alpha} \left( Q^*(o, a) - V^*(o) \right) \right). \tag{35}$$

$\square$

**Theorem.**3.3 *Given Eq. (23), the optimal policy for each agent is derived, and consistency between the local optimal policies $\pi_i^*$ and the global optimal policy $\pi_{tot}^*$ is guaranteed. This consistency holds for individual $\alpha_i$ assignments across agents.*

*Proof.* To provide the proof, we initially return to the decomposition framework of the Q-function in MARL setting, which is:

$$Q(o, a) = Q^r_{tot}(o, a) + \sum_{i=1}^{N} \alpha_i \cdot Q^{c,i}(o, a). \tag{36}$$

In this decomposition framework, the global $Q$ is divided into two parts: $Q^r_{tot}$ represents the computation of the return, and $Q^{c,i}$ serves as the global mapping of each agent's conservatism level.

Consider a global perspective that a global $\alpha_{tot}$ controls the whole conservatism level:

$$Q(o, a) = Q^r_{tot}(o, a) + \alpha_{tot} \cdot Q^c_{tot}(o, a). \tag{37}$$

Compare Eq.(36) with Eq.(37), the computation of the return is the same, while the deviation part varies due to the conservatism level. These two equations implicitly indicate that:

$$\alpha_{tot} \cdot Q^c_{tot}(o, a) = \sum_{i=1}^{N} \alpha_i \cdot Q^{c,i}(o, a). \tag{38}$$

Back to the definition of $Q^{c,i}$ and $Q^c_{tot}$, According to Eq.(38), we have:

$$\alpha_{tot} \cdot \log\left(\frac{\pi^*_{tot}(a \mid o)}{\pi_b(a \mid o)}\right) = \sum_i \alpha_i \cdot \log\left(\frac{\pi^*_i(a_i \mid o_i)}{\pi_b(a_i \mid o_i)}\right). \tag{39}$$

Then we separate the parts involving Q-function and V-function from the parts involving $\pi$ in Eq.(26):

$$\frac{\pi^*_{tot}(a \mid o)}{\pi_b(a \mid o)} = \exp\left(\frac{1}{\alpha_{tot}}(Q^*(o, a) - V^*(o))\right), \tag{40}$$

$$\log\left(\frac{\pi^*_{tot}(a \mid o)}{\pi_b(a \mid o)}\right) = \frac{1}{\alpha_{tot}}(Q^*(o, a) - V^*(o)), \tag{41}$$

$$\alpha_{tot} \cdot \log\left(\frac{\pi^*_{tot}(a \mid o)}{\pi_b(a \mid o)}\right) = Q^*(o, a) - V^*(o). \tag{42}$$

Similarly, the local parts in Eq.(23) can be written as :

$$\alpha_i \cdot \log\left(\frac{\pi^*_i(a_i \mid o_i)}{\pi_b(a_i \mid o_i)}\right) = w^r_i(\mathbf{o})\left(Q^{r^*}_i(o_i, a_i) - V^{r^*}_i(o_i)\right)$$
$$+ \alpha_i \cdot \left(Q^{c,i^*}(\mathbf{o}, a) - V^{c,i^*}(\mathbf{o})\right). \tag{43}$$

With Eq.(13) - (16), we can sum both sides of Eq.(43) with respect to $i$:

$$\sum_i \alpha_i \cdot \log\left(\frac{\pi^*_i(a_i \mid o_i)}{\pi_b(a_i \mid o_i)}\right) = \sum_i w^r_i(\mathbf{o})\left(Q^{r^*}_i(o_i, a_i) - V^{r^*}_i(o_i)\right)$$
$$+ \sum_i \alpha_i \cdot \left(Q^{c,i^*}(\mathbf{o}, a) - V^{c,i^*}(\mathbf{o})\right), \tag{44}$$

$$\alpha_{tot} \cdot \log\left(\frac{\pi^*_{tot}(a \mid o)}{\pi_b(a \mid o)}\right) = \sum_i w^r_i(\mathbf{o})\left(Q^{r^*}_i(o_i, a_i) - V^{r^*}_i(o_i)\right)$$
$$+ \sum_i \alpha_i \cdot \left(Q^{c,i^*}(\mathbf{o}, a) - V^{c,i^*}(\mathbf{o})\right), \tag{45}$$

$$\alpha_{tot} \cdot \log\left(\frac{\pi^*_{tot}(a \mid o)}{\pi_b(a \mid o)}\right) = Q^*(o, a) - V^*(o). \tag{46}$$

The transformation from Eq.(44) to Eq.(46) implies that with Eq.(23):

$$\pi_{tot}^*(a \mid o) = \prod_{i=1}^{N} \pi_i^*(a_i \mid o_i),$$

which means Eq.(23) not only allows for the derivation of the optimal policy for each agent, but also ensures consistency between the local optimal policies $\pi_i^*$ and the global optimal policy $\pi_{tot}^*$, even when each agent has a distinct $\alpha_i$.

$\square$

**Proposition.**3.4 *From Eq. (24), each agent's conservatism-based value function $V_i^c$ is updated through the following optimization:*

$$\min_{V_i^c} \mathbb{E}_{(o_i,a_i) \sim \mathcal{D}} \left[ \exp \left( \frac{w_i^r(\mathbf{o})}{\alpha_i} \left( Q_i^r\left(o_i,a_i\right) - V_i^r\left(o_i\right) \right) \right. \right.$$
$$\left. \left. + \left( Q^{c,i}\left(o,a\right) - V^{c,i}\left(o\right) \right) \right) + \frac{w_i^r(\mathbf{o})V_i^r\left(o_i\right) + \alpha_i V^{c,i}\left(o\right)}{\alpha_i} \right]. \tag{47}$$

*Proof.* The proof (similar to (Wang et al., 2024)) follows by showing that the first-order optimal condition of the above optimization objective, where the derivative with respect to $V^{c,i}$ equals zero, is exactly the Eq.(24):

$$\frac{\partial}{\partial V^{c,i}(o)} \left[ \exp \left( \frac{w_i^r(\mathbf{o})}{\alpha_i} \left( Q_i^r\left(o_i,a_i\right) - V_i^r\left(o_i\right) \right) \right. \right.$$
$$\left. \left. + \left( Q^{c,i}\left(o,a\right) - V^{c,i}\left(o\right) \right) \right) + \frac{w_i^r(\mathbf{o})V_i^r\left(o_i\right) + \alpha_i V^{c,i}\left(o\right)}{\alpha_i} \right] = 0 \tag{48}$$

$$\implies$$

$$\mathbb{E}_{a_i \sim \pi_b} \left[ -\exp \left( \frac{w_i^r(\mathbf{o})}{\alpha_i} \left( Q_i^r(o_i,a_i) - V_i^r(o_i) \right) \right. \right.$$
$$\left. \left. + \left( Q^{c,i}(o,a) - V^{c,i}(o) \right) \right) + 1 \right] = 0. \tag{49}$$

From the perspective of seeking the optimal function, we can have:

$$\mathbb{E}_{a_i \sim \pi_b} \left[ \exp \left( \frac{w_i^r(\mathbf{o})}{\alpha_i} \left( Q_i^{r^*}(o_i,a_i) - V_i^{r^*}(o_i) \right) \right. \right.$$
$$\left. \left. + \left( Q^{c,i^*}(\mathbf{o},a) - V^{c,i^*}(\mathbf{o}) \right) \right) \right] = 1. \tag{50}$$

This result implies that the optimal form of $V_i^c$ can be obtained by solving the convex optimization problem in Eq.(47).

$\square$

## D  EXPERIMENT SETTINGS

We select **Multi-Agent MuJoCo**(de Witt et al., 2020) and the **StarCraft Multi-Agent Challenge (SMAC)**(Samvelyan et al., 2019) as our experimental environments.

Multi-Agent MuJoCo, a benchmark for continuous multi-agent robotic control, is built on the MuJoCo environment. The Multi-Agent MuJoCo dataset we use was collected using the HAPPO(Kuba et al.,

2021) algorithm by (Wang et al., 2024) which contains four quality levels: expert, medium, medium-replay and medium-expert. The expert dataset is generated by employing the converged HAPPO algorithm, which involves training the algorithm until it reaches a state of convergence, where the agents have learned optimal policies. The medium dataset is generated by first training a policy online using HAPPO, early-stopping the training, and collecting samples from this partially-trained policy. The medium-replay dataset consists of recording all samples in the replay buffer observed during training until the policy reaches the medium level of performance. The medium-expert dataset is constructed by mixing equal amounts of expert demonstrations and suboptimal data. For all datasets, the hyperparameter *env_args.agent_obsk* is set to 1. The average returns of the datasets are listed in Table 2.

Table 2: The multi-agent MuJoCo datasets.

| Scenario | Quality | Average Return |
|----------|---------|----------------|
| 2-Agent Ant | expert | 2055.07 |
| 2-Agent Ant | medium | 1418.70 |
| 2-Agent Ant | medium-expert | 1736.88 |
| 2-Agent Ant | medium-replay | 1029.51 |
| 3-Agent Hopper | expert | 2452.02 |
| 3-Agent Hopper | medium | 723.57 |
| 3-Agent Hopper | medium-expert | 1190.61 |
| 3-Agent Hopper | medium-replay | 746.42 |
| 6-Agent HalfCheetah | expert | 2785.10 |
| 6-Agent HalfCheetah | medium | 1425.66 |
| 6-Agent HalfCheetah | medium-expert | 2105.38 |
| 6-Agent HalfCheetah | medium-replay | 655.76 |

The second environment, SMAC, is a widely-used benchmark for evaluating cooperative MARL methods. SMAC consists of a set of StarCraft II micro scenarios, and all scenarios are confrontations between two groups of units. Agents based on the MARL algorithm control the first group's units, while a built-in heuristic game AI bot with different difficulties controls the second group's units. Scenarios vary in terms of the initial location, number and type of units, and elevated or impassable terrain. The available actions for each agent include no operation, move[direction], attack [enemy id], and stop. The reward that each agent receives is the same. The hit-point damage dealt and received determines the agents' share of the reward. The offline SMAC dataset is collected by (Meng et al., 2021), using online-trained MAPPO(Kuba et al., 2021) agents. This is the largest publicly available dataset for SMAC and includes three quality levels: good, medium, and poor. We focus on three representative battle maps in our experiments: one hard map (5m_vs_6m) and two super hard maps (6h_vs_8z and corridor). The task types of the maps are listed in the Table 3. For each dataset in a map, we randomly sample 1000 episodes as our dataset. The average returns of SMAC datasets are listed in Table 4.

Table 3: SMAC maps for experiments.

| Map Name | Type |
|----------|------|
| 5m_vs_6m | homogeneous & asymmetric |
| 6h_vs_8z | micro-trick: focus fire |
| corridor | micro-trick: wall off |

# E  IMPLEMENTATION DETAILS

## E.1  ALGORITHM SUMMARY

In this section, we will give an explanation of the pseudocode for OMCDA. The pseudocode is shown in Algorithm. 1 We initialize the **behavior policy** $\pi_b$ through behavior cloning (Michie et al.,

Table 4: The SMAC datasets.

| Map Name | Quality | Average Return |
|----------|---------|----------------|
| 5m_vs_6m | good | 20.00 |
| 5m_vs_6m | medium | 11.03 |
| 5m_vs_6m | poor | 8.50 |
| 6h_vs_8z | good | 17.84 |
| 6h_vs_8z | medium | 11.96 |
| 6h_vs_8z | poor | 9.12 |
| corridor | good | 19.88 |
| corridor | medium | 13.07 |
| corridor | poor | 4.93 |

---

**Algorithm 1** Pseudocode of OMCDA

---

**Input:** Offline dataset $D$, $d_{tot}$
Initialize return-based state-value network $V_i^r$, constraint-based state-value network $V_i^c$, return-based action-value network $Q_i^r$, constraint-based action-value network $Q_i^c$, conservatism level $\alpha_i$, and policy network $\pi_i$ for agent $i = 1, 2, \ldots, n$.
**for** $t = 1$ **to** max-step **do**
    Sample batch transitions $(o, a, r, o')$ from $D$.
    Update return-based state-value function $V_i^r(o)$ for each agent $i$, via Eq. (51).
    Update constraint-based state-value function $V_i^c(o)$ for each agent $i$, via Eq. (25).
    Compute $V_{\text{tot}}^r(o')$ and $Q_{\text{tot}}^r(o, a)$, via Eq. (15) and Eq. (13).
    Update return-based action-value network $Q_i^r(o, a)$, via Eq. (53).
    Update constraint-based action-value network $Q_i^c(o, a)$, via Eq. (52).
    Update local policy network $\pi_i$ for each agent $i$, via Eq. (54).
    Calculate $m_i$ with Eq.(19) and update each agent's conservative degree $d_i$, via Eq. (17).
    Update each agent's conservatism level $\alpha_i$, via Eq. (55).
**end for**

---

1990) using the offline dataset. The procedure begins by initializing all necessary networks and parameters for each agent. At each iteration, the algorithm samples transitions from the dataset $D$ and performs sequential updates of both local and global networks.

**1. State-Value Updates:** The state-value functions $V_i^r$ and $V_i^c$ are updated first. Inspired by IQL(Kostrikov et al., 2021b) , we can implicitly update $V_i^r$ by leveraging the expectile loss, thus avoiding the use of out-of-distribution data. $V_i^r$ and $V_i^c$ are updated as following:

**Update $V_i^r$:** The return-based state-value function $V_i^r(o)$ for each agent is updated by minimizing the following objective:

$$\min_{V_i^r} \mathbb{E}_{(o_i, a_i) \sim \mathcal{D}} \left[ L_2^\tau \left( Q_i^r \left( o_i, a_i \right) - V_i^r \left( o_i \right) \right) \right], \tag{51}$$

where $L_2^\tau$ denotes the expectile loss with parameter $\tau$, balancing the updates based on the agent's value estimation errors.

**Update $V_i^c$:** The constraint-based state-value function $V_i^c(o)$ is updated using Eq.(25).

**2. Global Value Computation:** In this step, we compute the global term for return-based function. Here $V_{\text{tot}}^r(o')$ and $Q_{\text{tot}}^r(o, a)$ are calculated in Eq.(15) and Eq.(13).

**3. Action-Value Updates:** Each agent's action-value networks $Q_i^r$ and $Q_i^c$ are then updated. This step ensures that the agents maintain the correct mapping between their actions and the expected return as well as conservatism constraints.

**Update $Q_i^c$:** The constraint-based action-value function $Q_i^c$, along with the weight $w^{c,i}$ and bias $b^{c,i}$, is updated by minimizing the following objective, while $Q^{c,i}$ and $V^{c,i}$ are from Eq.(14) and Eq.(16):

$$\min_{\substack{Q_i^c, w^{c,i}, b^{c,i} \\ i=1,\cdots,n}} \mathbb{E}_{(\boldsymbol{o}, \boldsymbol{a}, \boldsymbol{o'}) \sim \mathcal{D}} \left[ \left( Q^{c,i}(\boldsymbol{o}, \boldsymbol{a}) - \gamma V^{c,i} \left( \boldsymbol{o'} \right) \right)^2 \right]. \tag{52}$$

**Update $Q_i^r$:** The return-based action-value function $Q_i^r$, weight $w_i^r$, and bias $b^r$ are updated according to the following minimization objective:

$$\min_{\substack{Q_i^r, w_i^r, b^r \\ i=1,\cdots,n}} \mathbb{E}_{(\boldsymbol{o},\boldsymbol{a},\boldsymbol{o}')\sim\mathcal{D}} \left[ \left( r(\boldsymbol{o},\boldsymbol{a}) + \gamma V_{tot}^r(\boldsymbol{o}') - Q_{tot}^r(\boldsymbol{o},\boldsymbol{a}) \right)^2 \right]. \tag{53}$$

**4. Policy Updates:** The agent's policy network is updated based on optimizing the following function.

**Update $\pi_i$:** The policy $\pi_i$ for each agent is updated by enforcing the KKT condition on Eq.(5) leveraging Eq.(22) :

$$\max_{\pi_i} \mathbb{E}_{(o_i,a_i)\sim\mathcal{D}} \left[ \exp\left( \frac{w_i^r(\boldsymbol{o})}{\alpha_i} \left( Q_i^r(o_i,a_i) - V_i^r(o_i) \right) \right. \right.$$
$$\left. \left. + \left( Q^{c,i}(o,a) - V^{c,i}(o) \right) \right) \cdot \log \pi_i\left( a_i \mid o_i \right) \right]. \tag{54}$$

**5. Conservatism Updates:** Finally, each agent's conservative degree $d_i$ is updated to ensure the balance between the risk and flexibility for each agent. After calculate $m_i$ with Eq.(19), we can update $d_i$ following Eq.(17). While the conservatism level $\alpha_i$ is adjusted to control the balance between deviation and conservatism.

**Update $\alpha_i$:** The conservatism level $\alpha_i$ is updated according to Eq.(55).

$$\min_{\alpha_i} \mathbb{E}_{(o_i,a_i)\sim\mathcal{D}} \left[ \alpha_i \cdot d_i - \alpha_i \cdot D_{\mathrm{KL}}\left( \pi_i \parallel \pi_b^i \right) \right]. \tag{55}$$

If the deviation from the behavior policy is less than $d_i$, $\alpha_i$ will decrease, allowing more flexibility for exploration. Conversely, if the deviation exceeds $d_i$, $\alpha_i$ will increase, pushing the policy to stay closer to the behavior policy.

### E.2    DETAILS OF OMCDA

The return-computation and constraint modules in the Q-function and V-function, and policy networks of OMCDA are represented by 3-layer ReLU activated MLPs with 256 units for each hidden layer. For the both weight networks of the two modules, we use 2-layer ReLU-activated MLPs with 64 units for each hidden layer. All the networks are optimized by Adam optimizer.

For the computation of the influence term, in practice, directly computing the derivatives can indeed lead to numerical instability. Therefore, we employ several techniques to stabilize the differentiation process: For continuous action spaces such as MuJoCo, we adopt a reparameterization method and use single-sample average for obtaining expectations for algorithm stability, simplifying the original differentiation process into a relationship between $Q^r$ and the log variance of policy network, which is then directly computed using deep learning libraries in PyTorch. For discrete action space environments like SMAC, due to the finite action set, we approximate the target derivative by applying small parameter perturbations to $\pi$ and using finite difference approximation trick.

In this paper, all experiments are implemented with Pytorch and executed on NVIDIA A100 GPUs.

### E.3    DETAILS OF BASELINES AND COMPARATIVE EVALUATION

We compare our approach with seven recent offline MARL algorithms: The multi-agent versions of BCQ(Fujimoto et al., 2019) and CQL(Kostrikov et al., 2021b) (referred to as BCQ-MA and CQL-MA), ICQ(Yang et al., 2021) , OMAR(Pan et al., 2022), CFCQL (Shao et al., 2024), OMIGA(Wang et al., 2024), and ComaDICE (Bui et al., 2024). Both BCQ-MA and CQL-MA utilize a linear weighted value decomposition structure for the multi-agent setting, similar to Eq. (1).

Table 6 and Table 7 presents the mean and standard deviation of average returns for the offline Multi-Agent MuJoCo and SMAC tasks with 5 random seeds. In these multi-agent scenarios, the complexity of the environment makes it challenging to assign conservative degree to individual agents,

as different agents' deviations from their behavior policies have varying impacts on the environment, which in turn influences the learning process. The dynamic conservative degree allocation mechanism in OMCDA assigns different conservatism levels to each agent based on their varying impacts on the system, which leads to better overall system performance. Moreover, by separating the return optimization from policy deviation management, OMCDA provides a more refined learning process, resulting in improved stability and effectiveness, enabling better collaboration and more efficient policy learning compared to other offline MARL methods.

Table 5: Hyper-parameter of OMCDA.

| Hyperparameter | Value |
|---|---|
| **OMCDA** | |
| Value network for return learning rate | 2e-4 |
| Value network for constraint learning rate | 4e-5 |
| Alpha learning rate | 1e-5 |
| Policy network learning rate | 2e-4 |
| Optimizer | Adam |
| Target update rate | 0.005 |
| Batch size | 128 |
| Discount factor | 0.99 |
| Hidden dimension | 256 |
| Expectile parameter $\tau$ | 0.7 |
| Initial conservative degree $d_i$ | 0.05 or 0.1 or 0.2 |

### E.4 HYPERPARAMETERS

For multi-agent MuJoCo and SMAC, the hyperparameters of OMCDA are listed in Table 5. Since we aim to quickly learn the return while maintaining stability in deviation, we use different learning rates for the value network: one for return and another for the constraint, set to $2 \times 10^{-4}$ and $4 \times 10^{-5}$, respectively. In OMCDA, the conservative degree $d$ is an important parameter. When the value of $d$ is large, the algorithm's overall conservative degree increases, providing the system with greater flexibility in policy exploration. Conversely, when the conservative degree is smaller, the policy tends to align more closely with the behavior policy. In the multi-agent MuJoCo environment, for the expert dataset, we set the initial $d_i = 0.05$ for each agent to guarantee effective regularization, while for other datasets, we set the initial $d_i = 0.2$ to maintain moderate deviation. In the SMAC environment, for the good dataset, we set the initial $d_i = 0.1$ for each agent, and $d_i = 0.2$ for the other datasets.

### E.5 DETAILS OF EQ. (5)

Consider a common case where there's only a global constraint:

$$\max_\pi E\left[\sum_{t=0}^\infty \gamma^t r_t\right], \quad \text{s.t.} \ E_{a \sim \pi}\left[\log \frac{\pi_t(a_t|o_t)}{\pi_b(a_t|o_t)}\right] \le d_{tot}. \tag{56}$$

Then according to the Lagrangian relaxation, the global conservatism level $\alpha_{tot}$ can be assigned and the Q-function is formulated as:

$$Q(o,a) = \mathbb{E}\left[\sum_{t=0}^\infty \gamma^t \left(r_t - \alpha_{tot} \cdot D_{\text{KL}}(\pi_t \parallel \pi_b)\right)\right]. \tag{57}$$

When considering Eq. (56) under the constraints specified in Eq. (5) and Proposition 3.2, the global constraint can be transformed step by step:

$$E_{a \sim \pi}\left[\log \frac{\pi_t(a_t \mid o_t)}{\pi_b(a_t \mid o_t)}\right] \le d_{tot}, \tag{58}$$

Table 6: Offline Multi-agent MuJoCo Tasks

| | | Multi-agent MuJoCo | | | | | | | |
|---|---|---|---|---|---|---|---|---|---|
| Task | Dataset | BCQ-MA | CQL-MA | ICQ | OMAR | CFCQL | OMIGA | ComaDICE | OMCDA |
| Hopper | expert | 77.85±58.04 | 159.14±313.83 | 754.74±806.28 | 2.36±1.46 | 802.33±544.89 | 859.63±709.47 | **2827.7±62.9** | 1214.25±830.72 |
| Hopper | medium | 44.58±20.62 | 401.27±199.88 | 501.79±14.03 | 21.34±24.90 | 389.75±105.67 | 1189.26±544.30 | 822.6±66.2 | **1441.53±488.91** |
| Hopper | m-replay | 26.53±24.04 | 31.37±15.16 | 195.39±103.61 | 3.30±3.22 | 567.54±453.65 | 774.18±494.27 | 906.3±242.1 | **1733.27±379.71** |
| Hopper | m-expert | 54.31±23.66 | 64.82±123.31 | 355.44±373.86 | 1.44±0.86 | 721.23±342.56 | 709.00±595.66 | **1362.4±522.9** | 1047.13±523.67 |
| Ant | expert | 1317.73±286.28 | 1042.39±2021.65 | 2050.00±11.86 | 312.54±297.48 | 1987.98±34.65 | 2055.46±1.58 | 2056.9±5.9 | **2056.95±6.43** |
| Ant | medium | 1059.60±91.22 | 533.90±1766.42 | 1412.41±10.93 | -1710.04±1588.98 | 1406.56±123.59 | 1418.44±5.36 | **1425.0±2.9** | 1376.03±141.55 |
| Ant | m-replay | 950.77±48.76 | 234.62±1618.28 | 1016.68±53.51 | -2014.20±844.68 | 854.21±128.98 | 1105.13±88.87 | 1122.9±61.0 | **1142.59±75.15** |
| Ant | m-expert | 1020.89±242.74 | 800.22±1621.52 | 1590.18±85.61 | -2992.80±6.95 | 978.87±65.45 | 1720.33±110.63 | 1813.9±68.4 | **1988.09±41.49** |
| HalfCheetah | expert | 2992.71±629.65 | 1189.54±1034.49 | 2955.94±459.19 | -206.73±161.12 | 2399.12±345.65 | 3383.61±552.67 | 4082.9±45.7 | **4963.92±126.69** |
| HalfCheetah | medium | 2590.47±110.35 | 1011.35±1016.94 | 2549.27±96.34 | -265.68±146.98 | 1845.43±76.78 | 3608.13±237.37 | 2664.7±54.2 | **3883.60±93.43** |
| HalfCheetah | m-replay | -333.64±152.06 | 1998.67±693.92 | 1922.42±612.87 | -235.42±154.89 | 1766.45±659.78 | 2504.70±83.47 | 2855.0±242.2 | **2993.03±271.84** |
| HalfCheetah | m-expert | 3543.70±780.89 | 1194.23±1081.06 | 2839.93±924.02 | -253.84±63.94 | 1934.23±867.43 | 2948.46±518.89 | 3889.7±81.6 | **4483.76±268.71** |

$$\sum_i E_{a\sim\pi}\left[\log\frac{\pi_t^i(a_t^i \mid o_t^i)}{\pi_b(a_t^i \mid o_t^i)}\right] \leq \sum_i d_i, \tag{59}$$

$$E_{a\sim\pi}\left[\log\frac{\pi_t^i(a_t^i \mid o_t^i)}{\pi_b(a_t^i \mid o_t^i)}\right] \leq d_i, \quad \sum_i d_i = d_{tot}, \ \forall i = 1\cdots n. \tag{60}$$

Therefore, we effectively achieve an equivalent transformation from global to local policy constraints. By comparing the Q-function under the global constraint in Eq. (57) with that under local constraints in Eq. (6), and noting that $E_{a\sim\pi}\left[\log\frac{\pi_t^i(a_t^i|o_t^i)}{\pi_b(a_t^i|o_t^i)}\right] = E_{a_i\sim\pi_i}\left[\log\frac{\pi_t^i(a_t^i|o_t^i)}{\pi_b(a_t^i|o_t^i)}\right]$, the following conclusion can be derived from this equivalence:

$$Q(o,a) = \mathbb{E}\left[\sum_{t=0}^{\infty}\gamma^t\left(r_t - \alpha_{tot}\cdot D_{\text{KL}}(\pi_t \parallel \pi_b)\right)\right] = \mathbb{E}\left[\sum_{t=0}^{\infty}\gamma^t\left(r_t - \sum_i \alpha_i\cdot D_{\text{KL}}(\pi_t^i \parallel \pi_b)\right)\right], \tag{61}$$

$$\alpha_{tot}\cdot\log\left(\frac{\pi(a\mid o)}{\pi_b(a\mid o)}\right) = \sum_i \alpha_i\cdot\log\left(\frac{\pi_i(a_i\mid o_i)}{\pi_b(a_i\mid o_i)}\right). \tag{62}$$

Eq. (62) effectively decomposes the global deviation term into local components, thereby establishing the foundation for both generating Eq. (38) and the deviation term in Eq. (12).

Table 7: Offline SMAC Tasks

| | | SMAC | | | | | | | |
|---|---|---|---|---|---|---|---|---|---|
| Task | Dataset | BCQ-MA | CQL-MA | ICQ | OMAR | CFCQL | OMIGA | ComaDICE | OMCDA |
| 5m_vs_6m | good | 7.76±0.15 | 8.08±0.21 | 7.87±0.30 | 7.40±0.63 | 8.13±0.32 | 8.25±0.37 | 8.7±0.5 | **10.47±0.24** |
| 5m_vs_6m | medium | 7.58±0.10 | 7.78±0.10 | 7.77±0.30 | 7.08±0.51 | 7.55±0.36 | 7.92±0.57 | 8.7±0.4 | **10.19±0.15** |
| 5m_vs_6m | poor | 7.61±0.36 | 7.43±0.10 | 7.26±0.19 | 7.27±0.42 | 7.49±0.12 | 7.52±0.21 | 8.1±0.5 | **9.57±0.18** |
| corridor | good | 15.24±1.21 | 5.22±0.81 | 15.54±1.12 | 6.74±0.69 | 14.25±0.78 | 15.88±0.89 | **18.0±0.1** | 14.72±0.60 |
| corridor | medium | 10.82±0.92 | 7.04±0.66 | 11.30±1.57 | 7.26±0.71 | 11.44±1.32 | 11.66±1.30 | 12.9±0.6 | **13.06±0.71** |
| corridor | poor | 4.47±0.94 | 4.08±0.60 | 4.47±0.43 | 4.28±0.49 | 4.89±0.37 | 5.61±0.35 | 6.4±0.5 | **6.54±0.51** |
| 6h_vs_8z | good | 12.19±0.23 | 10.44±0.20 | 11.81±0.12 | 9.85±0.28 | 11.87±1.25 | 12.54±0.21 | 13.1±0.5 | **14.14±0.21** |
| 6h_vs_8z | medium | 11.77±0.16 | 11.29±0.29 | 11.13±0.33 | 10.36±0.16 | 12.25±0.43 | 12.19±0.22 | 12.8±0.2 | **13.65±0.31** |
| 6h_vs_8z | poor | 10.84±0.16 | 10.81±0.52 | 10.55±0.10 | 10.63±0.25 | 10.89±0.47 | 11.31±0.19 | **11.4±0.6** | 10.83±0.10 |

# F ADDITIONAL RESULTS

## F.1 ANALYSIS ON THE COMPONENTS OF OMCDA

To analyze the solution to conservative degree allocation and assess the impact of Q-function decomposition in OMCDA, we conduct three distinct ablation studies: OMCDA-w/o-CDA, OMCDA-w/o-dq, and OMCDA-rd. In OMCDA-w/o-CDA, all agents are assigned the same conservative degree $d_i$, without the implementation of conservative degree allocation. In contrast, OMCDA-w/o-dq maintains the dynamic conservative degree allocation but eliminates the Q-function decomposition, preventing the separation of return optimization from policy deviation handling. OMCDA-rd introduces random allocation of the conservatism constraint, assigning each agent a random $d_i$, in which we hope to evaluate the importance of strategically assigning conservatism levels based on each agent's impact.

Experiments are conducted on the HalfCheetah and 6h_vs_8z tasks in the Multi-agent MuJoCo and SMAC environments, respectively. Figure. 5 shows that OMCDA consistently outperforms all

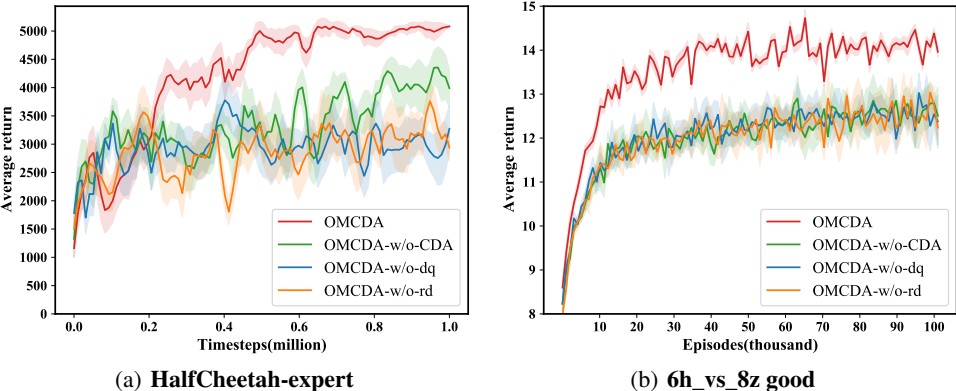

(a) **HalfCheetah-expert**  (b) **6h_vs_8z good**

Figure 5: Analysis on the components of OMCDA

ablated versions across both tasks. In OMCDA-w/o-CDA, the absence of dynamic conservative degree allocation results in imbalanced agent behavior, with some agents being overly conservative and others overly aggressive, leading to performance degradation. OMCDA-w/o-dq exhibits weaker results due to the entanglement of return maximization and constraint handling, which complicates learning and produces suboptimal policies. OMCDA-rd, which applies random conservatism allocation, demonstrates inferior performance, as randomly assigned $d_i$ values fail to account for each agent's unique influence on system performance.

These results confirm that both the dynamic conservative degree allocation and Q-function decomposition are essential for achieving better collaboration and learning efficiency in offline multi-agent environments, while the strategic assignment of conservatism is crucial for optimizing system performance.

## F.2 ANALYSIS ON THE TOTAL CONSERVATIVE DEGREE

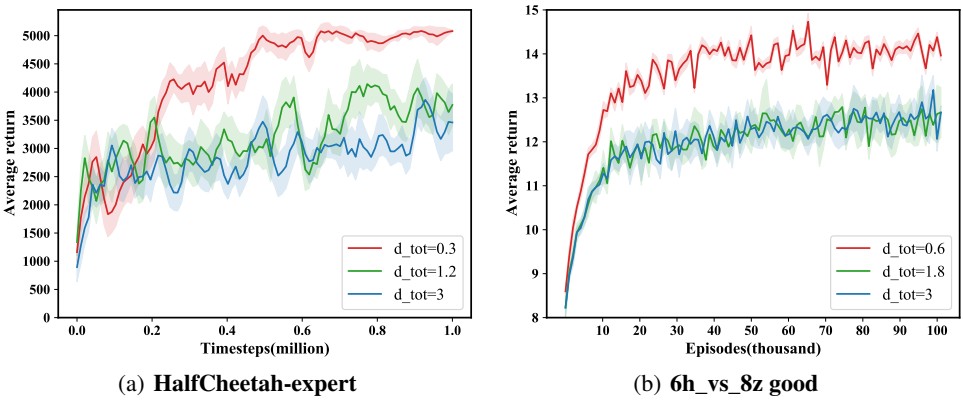

(a) **HalfCheetah-expert**  (b) **6h_vs_8z good**

Figure 6: Analysis on the total conservative degree

The total conservative degree $d_{tot}$ controls how much the system is permitted to deviate from the behavior policy. It establishes the total permissible deviation for the entire system, ensuring that agents do not diverge excessively from the behavior policy, helping to avoid the introduction of suboptimal actions into the system.

In our experiments on the HalfCheetah and 6h_vs_8z environments, which are based on high-quality datasets, a smaller $d_{tot}$ outperforms the other settings. This is because, in such environments, it is essential for the policies to stay closer to the behavior policy for stable performance. At the same time, the dynamic allocation of $d_i$ allows agents that have a significant impact on the system's return

to have some flexibility to deviate, while requiring other agents to remain more conservative and adhere closely to the behavior policy. The experimental results in Figure. 6 demonstrate that properly adjusting $d_{tot}$ improves system performance and allows influential agents to achieve beneficial deviations while maintaining overall system stability.

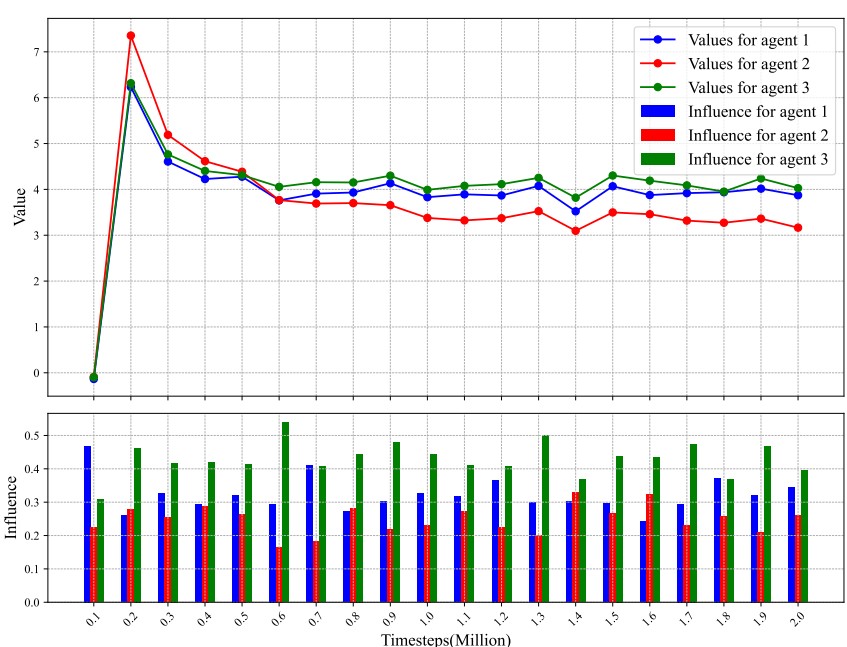

Figure 7: Analysis on the Influence Term on Hopper-medium-replay

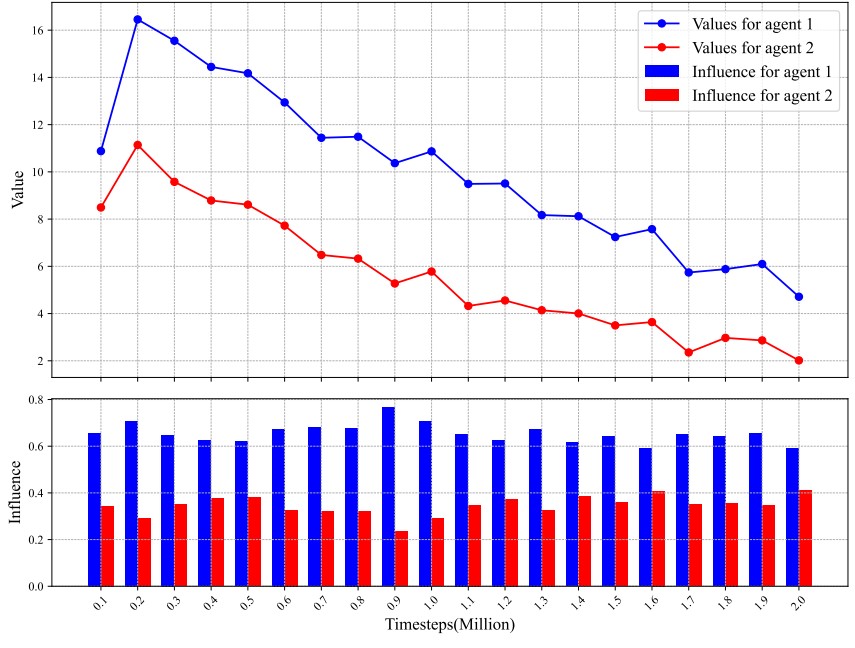

Figure 8: Analysis on the Influence Term on Ant-medium-expert

### F.3 ANALYSIS ON THE INFLUENCE TERM

In OMCDA, the influence of each agent on the system is the core metric for allocating conservatism levels. To determine whether we have accurately assessed each agent's influence in the environment, we conduct experiments to analyze the relationship between the computed influence of each agent and its corresponding return.

The results in Figure. 7 and Figure. 8 demonstrate that agents with higher $V_i^r$, representing higher individual returns, tend to be allocated more influence within the system, enabling them to have a stronger impact on system-wide performance. This proportional allocation allows high-return agents to further contribute to global objectives and optimize overall system behaviour.

By adjusting conservatism levels properly, OMCDA enhances individual agent performance and maximizes collective system return, promoting balanced and efficient learning across all agents.

