# OpenReview forum: "Who Matters Matters: Agent-Specific Conservative Offline MARL"
_ICLR.cc/2026/Conference — ICLR 2026 Poster_

### Official Review · Reviewer_PtX1 · 2025-10-30

**Soundness:** 3
**Presentation:** 3
**Contribution:** 3
**Rating:** 6
**Confidence:** 3

**Summary:**

The paper addresses challenges in Offline Multi-Agent Reinforcement Learning (MARL), particularly the need for effective collaboration among heterogeneous agents using fixed datasets. It introduces a novel framework called OMCDA (Offline MARL with Conservative Degree Allocation) that dynamically adjusts the conservatism level for each agent based on their impact on overall system performance.

**Strengths:**

1. The idea of dynamically adjusting conservatism levels based on agent impact is a significant advancement in the field, addressing limitations of uniform approaches
2. The paper provides extensive experimental results, showcasing the effectiveness of OMCDA across various scenarios.
3. The paper is well-organized, with a logical flow from introduction to methodology, experiments, and conclusions.

**Weaknesses:**

The design of the method is overly complicated, and the rationale behind some aspects of the design has not been justified.

**Questions:**

1. For each agent's Q and conservative term, can a similar effect be achieved by mixing them through the same mixer network？
  2. Is $\omega_i^r(o)$ in Eq. 13, $\omega_j^{c,i}(o)$ in Eq. 14 strictly greater than 0?
  3. Is it feasible to use another mixer network to fit the computation of $m$ in Eq. 20?

---

> ### Author Response · Authors · 2025-11-19
>
> **1. The design of the method is overly complicated, and the rationale behind some aspects of the design has not been justified.**
>
> Thank you for your question about our framework. Although the design may seem complex at first glance, each component has been thoughtfully integrated and is logically motivated. As illustrated in Figure 2, the workflow proceeds coherently and progressively, starting with agent-level influence assessment, moving to conservative term computation, and concluding with credit assignment. Furthermore, the dynamic conservative budget allocation and the network updates are theoretically grounded. The derivation and role of each term are also explained in detail in the paper.
>
> **2. For each agent's Q and conservative term, can a similar effect be achieved by mixing them through the same mixer network?**
>
> Thank you for your insightful comment. Your suggestion of using a shared mixing network for the different Q-functions is indeed theoretically feasible. Certain feature representation techniques from role-based MARL could potentially be incorporated into the design of our two types of Q-functions or their weights to enable the use of a common mixing network.
>
> However, such an approach would likely increase the representational complexity and make the algorithm more difficult to analyze theoretically. Our current framework, with separate Q-functions and a mixing network, offers greater intuitiveness and is more straightforward to analyze. Investigating how to design a universal mixing network effectively remains an important direction for our future research.
>
> **3. Is $w^r_i$ in Eq. 13, $w_{j}^{c,i}$ in Eq. 14 strictly greater than 0?**
>
> Yes, our approach follows most value decomposition methods (e.g., QMIX) in this regard, where the mixing network weights are constrained to be non-negative. To enforce this constraint, we employ a linear network combined with an absolute value activation function to ensure all weight outputs remain non-negative.
>
> **4. Is it feasible to use another mixer network to fit the computation of $m$ in Eq. 20?**
>
> Thank you for raising this interesting point. Exploring the use of different networks to approximate $m$ is indeed a valuable direction.
> Potential alternatives could include attention-based approaches, such as transformer architectures.
> Our current design deliberately focuses on interpretable influence estimation through policy gradient analysis and KL divergence sensitivity, which directly captures the inherent characteristics of our target problem domain while also easy to analyze. We will consider exploring more diverse mixing network designs as part of our future work to further enhance the method's performance.

---

> > ### Comment · Reviewer_PtX1 · 2025-11-26
> >
> > Thank you for the reply and clarifications. I would like to maintain my score.

---

### Official Review · Reviewer_U5BW · 2025-11-01

**Soundness:** 3
**Presentation:** 3
**Contribution:** 3
**Rating:** 8
**Confidence:** 4

**Summary:**

This paper introduces OMCDA, an offline multi-agent reinforcement learning (MARL) framework that dynamically allocates conservative degrees to individual agents based on their estimated influence on system returns. The key innovation lies in disentangling the Q-function into return and deviation components, enabling agent-specific regularization. The proposed method is evaluated on Multi-Agent MuJoCo and SMAC benchmarks, showing consistent performance gains over existing offline MARL baselines.

**Strengths:**

Novel Problem Formulation: Identifies and addresses the underexplored issue of heterogeneous conservatism in offline MARL — a real and important problem.

Technical Innovation: The Q-function decomposition and influence-based conservative allocation are technically sound and well-motivated.
Strong Empirical Results: Outperforms 7 strong baselines across diverse tasks and dataset qualities, with thorough ablations validating each component.

Theoretical Justification: Provides derivations for global-to-local policy consistency under agent-specific conservatism levels, enhancing credibility.

**Weaknesses:**

Scalability Concerns: The influence term requires computing partial derivatives w.r.t. per-agent KL divergences, which may become prohibitive in large-N systems or high-dimensional action spaces.

Sensitivity to Total Budget: Performance is sensitive to the choice of total conservative degree $d_{tot}$, yet no principled method is provided for setting it across tasks.

**Questions:**

1. What is the computational overhead of estimating influence terms compared to standard offline MARL methods, especially for N > 10 agents?

2. At present the total conservative budget $d_{tot}$ is treated as a hyper-parameter and tuned manually for each task. Have the authors explored any data-driven or adaptive procedure to automatically infer $d_{tot}$ so that the algorithm can adjust the global deviation allowance without human intervention?

---

> ### Author Response · Authors · 2025-11-19
>
> **1.  For W1 and Q1. The influence term requires computing partial derivatives per-agent KL divergences, which may become prohibitive in large-N systems or high-dimensional action spaces. What is the computational overhead of estimating influence terms compared to standard offline MARL methods, especially for $N > 10$ agents?**
>
> Thank you for your feedback on scalability. We acknowledge that computing the KL term can indeed increase computational load in large-scale scenarios. To mitigate this as the number of agents grows and environments become more complex, we have provided several computation-reduction tricks in Appendix E.2, which help reduce the computational cost to some extent.
>
> Regarding computational overhead, in the SMAC environment, the runtime of our method increased from 8 hours to 10 hours compared to the baseline. Given the performance improvement achieved, we believe this increase in computation time is acceptable.
>
> We agree that validating the algorithm's performance in larger-scale environments is crucial. However, due to the lack of mainstream experimental environments and datasets supporting such scales, conducting experiments in environments with $N > 10$ remains part of our future work.
>
> **2. For W2 and Q2. Performance is sensitive to the choice of total conservative degree $d_{tot}$, yet no principled method is provided for setting it across tasks. Have the authors explored any data-driven or adaptive procedure to automatically infer $d_{tot}$ so that the algorithm can adjust the global deviation allowance without human intervention?**
>
> We appreciate your insightful suggestion on $d_{tot}$. Building upon established findings from prior works (e.g., OMIGA) that demonstrate the necessity of adapting conservatism levels to dataset quality, our approach introduces $d_{tot}$ as a global conservatism regulator. While we have conducted ablation studies in Fig.4 validating the impact of different $d_{tot}$ settings, our primary contribution focuses on the novel dynamic allocation mechanism that optimally distributes this total conservatism budget among heterogeneous agents. The broader question of automated dataset-quality-aware $d_{tot}$ configuration, while important, represents a distinct research direction that we plan to investigate in future work on cross-dataset generalization.

---

### Official Review · Reviewer_mivS · 2025-11-02

**Soundness:** 1
**Presentation:** 2
**Contribution:** 2
**Rating:** 2
**Confidence:** 4

**Summary:**

The authors introduce Offline MARL with Conservative Degree Allocation (OMCDA) which aims to control the degree of conservatism for each agent by decomposing the Q-values to return maximizing and conservatism terms. They further use value decomposition akin to QMIX to compute the degree of conservatism for each agent based on how the KL regularization term impacts the global $V$. Experimental results are provided for SMACv1 and MA-MuJoCo.

**Strengths:**

The  illustrative matrix game is somewhat intuitive. Under some fixed total conservative degree, agent A has a higher influence on total reward, so it should follow more risky behavior, and deviate from the dataset distribution (assuming the higher return action is also in the dataset support).

Also, updating $\alpha$ in Eq.55 is interesting since we often want less hyperparameter tuning in offline RL.

**Weaknesses:**

1. The observation function $O$ is missing from the Markov Game formulation.
1. The reward function $r$ is defined in terms of joint observations, but it should be based on state.
1. The example used to motivate the work is slightly confusing. For football, the risk-taking vs risk-averse behavior between positions is about taking actions that can lead to states with a high return (i.e. score a goal) but may have low probability of reaching that goal. This is somewhat orthogonal to the concept of conservatism in offline MARL, where risk-averse vs risk-taking is about whether to follow the dataset distribution or not.
1. $Q_i^r, Q_i^c,  w_i^r, w_i^c$ from Eq. 13 and 14 is not well motivated. It seems that additional decomposition assumptions are required on both the reward component and constraint component. There is no detail provided in appendix E.5 as the authors mention.
1. Related to above, the validity of Eq.23 is questionable. For instance $Q^{c, i, *}$ is not well-defined. If value decomposition is used, the authors should be more clear about how the IGM assumption is extended to the offline setting and how it is affected by the conservative regularization term.
1. Even with a correct characterization of the decomposed terms, value decomposition is already quite restrictive. For instance, [1]  showed that it cannot solve simple matrix games, and [2] analyzed the limitations of value decomposition in the offline case.
1. Eq. 18 computes the influence term depending on how the KL term influences return. However, there is still a mismatch because the denominator is in terms of individual actions while the numerator is defined for joint observations/actions. Thus, there is an implicit dependency with the other agents observations. This kind of influence term can be biased in more complex environments if there is no 1-1 mapping between individual and joint observations.
1. SMAC results are only provided for v1, but this has already been show to be an outdated benchmark due to the deterministic transitions, and the fact that it can be solved by some simple algorithms that don’t even take observations into consideration [3]. Recent offline MARL algorithms such as ComaDICE already use smacv2 as a benchmark. This significantly weakens the experimental results of this paper. Also, smacv2 results would help with addressing my previous point since it is inherently more partially observable.


[1] Revisiting Some Common Practices in Cooperative Multi-Agent Reinforcement Learning (Fu et, al. 2021)

[2] AlberDICE: Addressing Out-Of-Distribution Joint Actions in Offline Multi-Agent RL via Alternating Stationary Distribution Correction Estimation (Matsunaga


[3] SMACv2: An Improved Benchmark for Cooperative Multi-Agent Reinforcement Learning (Ellis et, al. NeurIPS 2023 Datasets and Benchmarks)

**Questions:**

1. While the illustrative example in Table 1 makes some sense, what happens if the payoffs for taking action A for agent 1 are [-100, 3 ] instead of [3, 3]?
2. CFCQL and OMIGA seem to be important baselines since they both consider a similar problem of applying individual conservatism. How did you tune their hyperparameters? Is it similarly tuned fairly for OMCDA?

---

> ### Author Response · Authors · 2025-11-19
>
> **1. The observation function is missing from the Markov Game formulation. The reward function is defined in terms of joint observations, but it should be based on state.**
>
> Regarding the definitions, we initially adopted those from prior works [1]. In line with your feedback to align with more conventional terminology, we have added the definition of $\mathcal{O}$ and revised the reward definition in the rebuttal revision version.
>
> [1] Offline Multi-Agent Reinforcement Learning with Implicit Global-to-Local Value Regularization.(Wang et, al. 2023)
>
> **2. The example used to motivate the work is slightly confusing.**
>
> We thank the reviewer for the comment. There appears to be a misunderstanding regarding the football example. Our intention in using this analogy was to illustrate that in an offline setting, heterogeneous agents (players in different roles), have varying impacts on the team when they deviate from the behavioral policy, which in soccer is represented by the default tactical template. The example was introduced not to contrast risk-taking with risk-averse behavior, such as deciding whether to attempt to score a goal, but to emphasize that deviations by different roles affect team performance in different ways, which is central to the motivation of our work.
>
> To clarify further, consider a hunting analogy: two people go deer hunting—one searches for deer while the other shoots. Typically, if the shooter misses, the deer escapes. In this scenario, the shooter’s behavior is more conservative, while the searcher can take higher-risk actions. If this analogy helps, we will include it in the revised manuscript to make the concept clearer.
>
> **3. $Q_i^r, Q_i^c, w_i^r, w_i^c$ from Eq. 13 and 14 is not well motivated.**
>
> The decomposition and specification of \( Q_i^r \), \( Q_i^c \), \( w_i^r \), and \( w_i^c \) follow a common CTDE paradigm and do not require additional assumptions. Moreover, our approach aligns with established offline MARL methods that utilize value decomposition, such as ICQ[1] and OMIGA[2]. These works have already demonstrated the rationale behind such a decomposition. The purpose of Appendix E.5 is specifically to illustrate the analysis of the relationship between Eq.(5) and the deviation term in Eq.(12).
>
> [1] Believe What You See: Implicit Constraint Approach for Offline Multi-Agent Reinforcement Learning.(Yang et, al. 2021)
>
> [2] Offline Multi-Agent Reinforcement Learning with Implicit Global-to-Local Value Regularization.(Wang et, al. 2023)
>
> **4. The authors should be more clear about how the IGM assumption is extended to the offline setting.**
>
> Our formulation in Eq.(23) is an extension of Eq.(22), essentially constituting a decomposition from the global optimal policy to local optimal policies. The rationale for employing such a decomposition form via value decomposition has been thoroughly established and discussed in prior work like OMIGA.
>
> The application of the IGM principle in the offline setting is grounded in **a well-established line of research in offline MARL**. Previous works, including OMAC[1], have provided a detailed analysis that justifies this extension. Furthermore, both OMAC and OMIGA employ a conservative regularization term in their methodologies. These works have already experimentally validated the soundness and effectiveness of using such a regularization term under the IGM principle.
>
> [1] Offline Multi-Agent Reinforcement Learning with Coupled Value Factorization.(Wang et, al. 2023)
>
> **5. Even with a correct characterization of the decomposed terms, value decomposition is already quite restrictive.**
>
> Value decomposition has been widely adopted in offline MARL(like OMIGA, OMAC). Its rationale and effectiveness have been robustly demonstrated through extensive experiments. Furthermore, empirical results from these works show that their performance is competitive with alternative approaches like AlberDICE, which tackles the problem from a different perspective.

---

> ### Author Response · Authors · 2025-11-19
>
> **6. Concern about the influence term.**
>
> Our influence term quantifies the impact of an individual policy's deviation on the global return. Therefore, the KL divergence uses individual observation and action, while the global value function uses joint ones. Moreover, as demonstrated in common offline MARL methods such as OMIGA and ICQ, the decomposition of a global policy into individual policies ($\pi_{tot}(a \mid o) = \prod_{i=1}^{N} \pi_i(a_i \mid o_i)$) ensures the rationality of analyzing the global impact from an individual perspective. By distributing global information to the individual level, this decomposition ensures the rationality of analyzing the global impact from an individual perspective.
>
> **7. SMAC results are only provided for v1.**
>
> Following common practice, our experiments used standard environments and datasets. While most offline MARL research uses SMAC V1, we performed additional tests on the more challenging SMAC V2 to address the reviewer's concern.
>
> Using the SMAC V2 dataset from [1], we evaluated our method against MAICQ, which was the top performer in that study [1]. Results confirm our approach remains highly effective in partially observable environments like SMAC V2.
>
> | SMAC V2 | MAICQ | OMCDA(ours) |
> |------------------------|---------------|----------|
> | terran\_5\_vs\_5       | 13.7 ± 1.7    |  13.9 ±  1.2     |
> | zerg\_5\_vs\_5         | 10.6 ± 0.7    |  12.9 ±  1.5     |
> | terran\_10\_vs\_10     | 14.4 ± 0.7    | 16.3 ± 2.3      |
>
> [1]Off-the-Grid MARL: Datasets with Baselines for Offline Multi-Agent Reinforcement Learning.(Formanek et, al. 2023)
>
> **8. While the illustrative example in Table 1 makes some sense, what happens if the payoffs for taking action A for agent 1 are [-100, 3 ] instead of [3, 3]?**
>
> When the values are adjusted to -100 and 3, the calculation of the team's expected payoff based on mixed strategies becomes more complex. Specifically, the update direction of the optimal strategy is conditioned on the initial behavior policy. For example, assuming both players start with an initial policy of (0.5, 0.5) and are allowed a total deviation of 0.3, the optimal strategies they can learn are (0.5, 0.5) and (0.2, 0.8) respectively. However, if the initial policies are (0.9, 0.1) and (0.2, 0.8), the resulting optimal strategies become (1, 0) and (0, 1) respectively. In our original example, the parameters were set to 3 and 3, which was designed to simplify the analysis and thereby facilitate reader comprehension.
>
> **9. CFCQL and OMIGA seem to be important baselines since they both consider a similar problem of applying individual conservatism. How did you tune their hyperparameters? Is it similarly tuned fairly for OMCDA?**
>
> For CFCQL, we adopted the parameter settings from its ablation study, which yielded the best performance under varying dataset quality levels. Accordingly, we selected the optimal parameter values of 0.3, 0.4, and 0.6 for expert, medium, and poor-quality datasets, respectively.
>
> It should be noted that OMIGA employs **a uniform level of conservatism** across all agents, rather than applying individualized conservatism. Since we used the same environment and dataset provided by OMIGA, we directly adopted their reported experimental results.

---

> > ### Comment · Reviewer_mivS · 2025-11-26
> >
> > Thank you for the rebuttal.
> >
> > However, a lot of my concerns/questions are not addressed directly.
> > In my opinion, "It was used in previous work" is not a good justification for addressing concerns.
> >
> > Take Weaknesses 4 and 5 as an example. For IGM in general, the implicit assumption is that the environment should have decomposable optimal Q-values (e.g. Chapter 9 in [1]). However, Eq. 13-14 are introducing additional decompositions for $Q^c$. Again, this is not explained in Appendix E.5 although it is written in the main text that it is (Line 244). It also seems that this decomposition is different from OMIGA. For instance, $V^c$ is not defined. Eq.14 is also now more confusing after a second look as $Q^{c,i}$ and $Q^c_j$ both have an agent index. Please elaborate in more detail why Weaknesses 4 and 5 can be justified (either by previous work or by the authors' own analysis).
> >
> > For W6, IGM is still a strong assumption, and the authors should be upfront about this in terms of limiting the scope of this work. Performing well enough on specific benchmarks is not a justification for universally accepting IGM as a justifiable approach, especially since many previous work do not assume this (e.g. HARL for online MARL or ComaDICE for offline MARL). In reality, it depends on the environment whether IGM is justified.
> > In particular, IGM is implicitly assuming that the optimal Q-value (unique in each environment) can be decomposed. This implies that the optimal policy is also always factorized. Intuitively,  this is saying that each agent does not need to consider the other agents' actions. There are many other environments or real-world MARL settings where this kind of assumption may not hold.
> >
> > Finally, I appreciate the addition of SMACv2 experiments. However, ICQ is a weaker baseline, so it is still unclear to me whether OMCDA does improve over OMIGA which is a more important baseline in terms of showing the benefit of agent-wise conservatism.
> >
> > [1] Albrecht, Stefano V., Filippos Christianos, and Lukas Schäfer. Multi-Agent Reinforcement Learning: Foundations and Modern Approaches. MIT Press (2024).​

---

> ### Author Response · Authors · 2025-12-02
> **(1/2)**
>
> Thank you for your feedback. To clarify the questions you raised, we provide a detailed discussion of W4 and W5 on this page, and of W6 on the next page, as follows.
>
> In this page, we provide a detailed explanation showing how our decomposition in Eq.(12), although differing in form from prior works such as OMIGA, remains aligned with them in its principled application of IGM within the offline setting. We further demonstrate why Eq.(13) and Eq.(14) are consistent with value decomposition principles and are theoretically well-grounded.
> The definitions of $V^{c}$ for the single-agent and multi-agent cases are provided in our Eq.(11) and Eq.(16), respectively.
>
> Our key motivation is to assign different conservative degrees to each agent. To achieve this, it is essential to understand how an individual agent’s deviation from the behavior policy influences the overall return. In offline RL with regularization, as seen in methods like OMIGA, both the Q-function and value function incorporate entangled components that blend the return and the constraint term. This entanglement complicates the assessment of an agent’s specific impact on the return.
>
> To address this, in Eq.(12), we decompose the joint Q-function into two parts: one that estimates the return, and another that accounts for the deviation constraint. Consequently, we obtain a joint reward Q-function $Q^{r}_{tot}$ with global information and $n$ joint constraint Q-functions $Q^{c,1}, \cdots, Q^{c,n}$, also with global information. Accordingly, we assign a separate mixing network to the joint function $Q^{r}\_{tot}$ and to each joint $Q^{c,i}$ function. Specifically, for Eq.(14), since each $Q^{c,i}$ utilizes a mixing network weighted by all agents, we use the index $j$ in Eq.(14) to sum over all agents, avoiding ambiguity with the index $i$.
>
> Furthermore, by comparing the optimal function expressions in our Eq.(22) and Eq.(23) with their counterparts in OMIGA, it becomes evident that our optimality formulation is essentially a decomposed, agent-specific instantiation of the same underlying optimum within the framework established by Eq.(12), (13), and (15). We have provided detailed proofs in Proposition 3.1, Proposition 3.4, Theorem 3.3, and Appendix C, which explain why consistency between the global and local optima is preserved in this agent-specific setting. Thus, under the joint influence of $Q^{r}_{tot}$ and $Q^{c,i}$ obtained via the mixing networks in Eq.(13) and (14), the optimality direction of the decoupled global Q-function in Eq.(12) is consistent with that of the non-decoupled global Q-function in OMIGA. This also ensures that the decompositions in Eq.(13) and (14) retain the same theoretical validity.
>
> We have supplemented the experiments with OMIGA on the corresponding SMAC V2 dataset. The results demonstrate that our method outperforms OMIGA on SMAC V2, indicating that our use of agent-specific conservative degrees effectively addresses the challenge of heterogeneous agent collaboration in offline settings.
>
> | SMAC V2 | OMIGA | OMCDA(ours) |
> |------------------------|---------------|----------|
> | terran\_5\_vs\_5       |  13.79 ±  1.7   |  13.9 ±  1.2     |
> | zerg\_5\_vs\_5         |  10.88 ±  1.1   |  12.9 ±  1.5     |
> | terran\_10\_vs\_10     | 15.43 ±  1.4   | 16.3 ± 2.3      |

---

> ### Author Response · Authors · 2025-12-02
> **(2/2)**
>
> We thank the reviewer for highlighting the theoretical limitations of value decomposition, particularly regarding the representation of non-monotonic payoff structures as discussed in [1]. We fully acknowledge that value decomposition methods (e.g., VDN/QMIX) cannot represent every possible joint value function.
>
> However, we believe this design choice is justified and necessary for our specific setting—Offline MARL on high-dimensional cooperative tasks, for the following three reasons:
>
> 1. Practicality in Target Benchmarks vs. Theoretical Worst-Cases:
> While [1] demonstrates failure cases in specific matrix games where exists
> symmetric optimal strategies because of multi-modal reward, our work focuses on complex, high-dimensional cooperative benchmarks (e.g., SMAC, SMAC-v2, Multi-Agent MuJoCo). In these tasks, the reward structures are generally well-aligned with the Individual-Global-Max (IGM) assumption.
> We are aware of works like ComaDiCE or AlberDICE that utilize linear programming to handle more complex, multi-modal reward landscapes theoretically. However, in the context of the standard benchmarks used in this paper, such extreme reward complexity is rarely the bottleneck. Instead, the primary challenge lies in high-dimensional coordination and stability. Empirically, value decomposition remains the dominant and most effective paradigm for these tasks due to its scalability and training efficiency.
>
> 2. Inductive Bias as Crucial Regularization in Offline Settings: In the offline setting, "expressiveness" is a double-edged sword. A fully expressive joint Q-function is highly susceptible to overfitting and overestimating the value of out-of-distribution (OOD) actions due to the lack of interactive exploration.
> Value decomposition imposes a structural inductive bias (e.g., monotonicity). In our offline context, this bias acts as a powerful implicit regularizer. It constrains the hypothesis space, preventing the learned Q-function from arbitrarily overestimating unobserved joint actions. This structural constraint significantly enhances the stability and generalization of the learned policy, which is often more critical than theoretical expressiveness in offline scenarios.
>
> 3. Focus on Conservatism Allocation and Credit Assignment among heterogeneous agents: Most importantly, we wish to clarify that our core contribution is orthogonal to the debate regarding the theoretical upper bounds of value decomposition. Our research does not aim to expand the representational capacity of decomposition; rather, we focus on the structural credit assignment problem under offline constraints for heterogeneous agents.
>
> Specifically, we address the question: "Given a decomposed structure, how do we optimally weigh agents to balance local performance and global conservatism?"
> - The Challenge of Heterogeneity: In heterogeneous teams (e.g., SMAC maps with mixed unit types), agents have varying vulnerabilities to distribution shift. Uniform conservatism (applying the same penalty to all) is suboptimal because it over-penalizes robust agents while potentially under-penalizing fragile ones.
> - Our Solution (Adaptive Conservatism): Our method introduces an adaptive weighting mechanism that dynamically allocates "conservatism." By assigning higher weights (importance) to agents that are more reliable or critical for the current state, and adjusting penalties based on individual uncertainty, we enhance coordination without altering the underlying decomposition architecture.
>
> And our experiments on heterogeneous benchmarks (specifically the challenging SMAC maps and heterogeneous Multi-Agent MuJoCo tasks) demonstrate that this allocation strategy significantly outperforms baselines that use uniform or static weighting. The results confirm that "correctly allocating conservatism" is as critical as the conservatism itself.
>
>
> Finally, We appreciate the reviewer’s insight regarding the theoretical landscape of MARL. To address this, we will revise our Related Work section to include a detailed discussion on LP-based and DICE-based methods, specifically ComaDiCE and AlberDICE, to clearly differentiate our research focus:
> 1. We appreciate the reviewer’s insight regarding the theoretical landscape of MARL. To address this, we have revised our Related Work section to include a detailed discussion on LP-based and DICE-based methods, specifically ComaDiCE and AlberDICE, to clearly differentiate our research focus:
>
> 2. Clarification of Our Scope: In contrast, our work is scoped to Offline MARL on high-dimensional cooperative benchmarks (e.g., SMAC, MuJoCo), where the reward functions are generally well-behaved. In this specific scope, we argue that the primary challenge is not the representational limit of the decomposition, but rather the distribution shift and overestimation inherent to offline learning.
>
> [1] Albrecht, Stefano V. Multi-Agent Reinforcement Learning: Foundations and Modern Approaches. MIT Press (2024).

---

### Meta-Review · Area_Chair_vQcR · 2026-01-05

**Summary:**

Strength: The paper studies a timely and relevant problem in offline MARL. The proposed method is well motivated, and appears effective with good empirical performance and theoretical justification.

Weakness: The method appears to be complicated with potential concerns on scalability and sensitivity aspects. The experimental results could also be strengthened.

**Reviewer Concerns:**

Weakness: The method appears to be complicated with potential concerns on scalability and sensitivity aspects. The experimental results could also be strengthened.

**Reviewer Scores:**

The positive reviewers will keep their scores. The negative reviewer might increase his/her score.

---

### Decision · Program_Chairs · 2026-01-26

Accept (Poster)